# MESH FIELD THEORY: PORT–HAMILTONIAN FORMULATION OF MESH-BASED PHYSICS

## ABSTRACT

We present *Mesh Field Theory (MeshFT)* and its neural realization, MeshFT-Net: a structure-preserving framework for mesh-based continuum physics that cleanly separates the physics' *topological* structure from its *metric* structure. Imposing minimal physical principles (locality, permutation equivariance, orientation covariance, and energy balance/dissipation inequality), we prove a reduction theorem for mesh-based physics. Under these conditions, the physical dynamics admit a local factorization into a port–Hamiltonian form: the conservative interconnection is fixed uniquely by mesh topology, whereas metric effects enter only through constitutive relations and dissipation. This reduction clarifies what must be fixed and what should be learned, directly informing MeshFT-Net's design. Across evaluations on analytic and realistic datasets, physics-consistency tests, and out-of-distribution validation, MeshFT-Net achieves near-zero energy drift and strong physical fidelity—correct dispersion and momentum conservation—along with robust extrapolation and high data efficiency. By eliminating non-physical degrees of freedom and learning only metric-dependent structure, *MeshFT* provides a principled inductive bias for stable, faithful, and data-efficient physical simulation.

## 1 INTRODUCTION

Machine learning is increasingly used to accelerate continuum simulations, from data-driven surrogates and weak-form training to operator learning that transfers across meshes and parameters (Lu et al., 2021; Kovachki et al., 2023; E & Yu, 2018; Li et al., 2021; Gupta et al., 2021; Raissi et al., 2019; Sirignano & Spiliopoulos, 2018; Ummenhofer et al., 2020; Cao, 2021; Battaglia et al., 2018). Another line of work represents a discretized domain as a graph built from a mesh and learns time evolution by message passing, yielding strong results in analysis of fluid and deformable solids while remaining flexible in resolution and topology (Pfaff et al., 2021; Sanchez-Gonzalez et al., 2020).

However, a key structural point is often left implicit. In exterior calculus on a manifold (Flanders, 1963), the exterior derivative $d$ is *topological*—metric-independent—while geometry and material properties appear only through metric-dependent operators such as the Hodge star $\star$. In almost all existing learned mesh simulators, these roles are conflated, letting non-physical effects contaminate predicted fields and produce spurious modes and instabilities.

Discrete exterior calculus (DEC) (Hirani, 2003; Desbrun et al., 2005a;b) argues that mesh topology already provides the algebraic backbone for differential operators describing mesh-based physics. Also, metric-dependent part is clearly divided from such topological structures. Classical structure-preserving numerical schemes—e.g., finite-difference time-domain (FDTD)—exploits the same geometric structure and, as a results, yield stable and conservative updates (Yee, 1966; Taflove et al., 2005; Noguchi et al., 2020; Bossavit, 1998). These observations suggest a clean division of roles in mesh-based physics simulators: hard-code physics' topological structures based on the algebraic backbone given by fixed mesh topology, and learn only metric-dependent structures.

We develop this idea as *Mesh Field Theory (MeshFT)* and its neural realization MeshFT-Net. We formalize four minimal physical requirements—locality, permutation equivariance, orientation covariance, and energy balance/dissipation inequality—and prove a reduction theorem. Under these conditions, the mesh-based physics admit a local reduction to port–Hamiltonian in which the conservative interconnection is uniquely fixed by mesh topology, while metric-dependent effects enter only through constitutive or dissipative operators. This reduction removes non-physical freedoms

and focuses learning on the metric-dependent parts. Guided by the theorem, MeshFT-Net is designed as a sparse and orientation-aware model that updates states with a symplectic conservative step with an explicit dissipative step, without partial differential equation (PDE) residual losses.

Across evaluations on analytic dataset and real acoustic-scattering data from *The Well* (Ohana et al., 2024; Mandli et al., 2016), physics-consistency tests, and out-of-distribution (OOD) validation, MeshFT-Net achieves near-zero energy drift with correct dispersion and momentum behavior, strong generalization, and higher data efficiency relative to baselines. These results strongly support the view that making the topology/metric separation explicit is an principled inductive bias for stable, faithful, and data-efficient mesh-based simulation.

Our contributions can be summarized as follows:

- **Reduction theorem.** We formalize four minimal physical requirements and prove a local reduction theorem for mesh-based physics: the dynamics reduce to a port–Hamiltonian form in which the conservative interconnection is *uniquely* fixed by mesh topology, while only metric-dependent constitutive and dissipative effects are learnable.

- **Neural architecture.** We design a neural mash-based physics simulator (MeshFT-Net) that fixes topological wiring and learns only metric maps for constitutive relation and dissipation.

- **Empirical results.** Across evaluations on analytic dataset and real acoustic-scattering data from *The Well*, physics-consistency tests, and OOD validation, MeshFT-Net shows near-zero energy drift with strong physical fidelity, robust extrapolation performance, and higher data efficiency relative to baselines.

## 2 RELATED WORKS

**MeshGraphNets(MGN).** MGN learn mesh-based physics via message passing on graphs induced by meshes (Pfaff et al., 2021). Nodes carry physical states (velocity, pressure, strain), while edges encode adjacency and geometry (coordinate differences, distances); features can include conditioning signals (boundary types, material parameters). A standard encoder–processor–decoder (Battaglia et al., 2018) applies $T$ permutation-equivariant message-passing steps $\bar{m}_i = \sum_{j \in \mathcal{N}(i)} \phi_e(h_i, h_j, e_{ij})$, $h'_i = \phi_v(h_i, \bar{m}_i; g)$, where $h_i, e_{ij}$ are node/edge features, $g$ optional global features, and $\phi_e, \phi_v$ shared learnable maps. A decoder outputs accelerations or state increments for time integration. MGN transfer across geometries and resolutions in fluids, deformable solids, and contact settings.

**Structure-Preserving Learning for Physical Dynamics.** Hamiltonian Neural Networks (HNN) learn a Hamiltonian $H_\theta$ to induce $X_{H_\theta}$ (in canonical coordinates, $X_{H_\theta}(q, p) = (\partial_p H_\theta, -\partial_q H_\theta)$); variants learn a Lagrangian $L_\theta$ (Greydanus et al., 2019; David & Méhats, 2023; Eidnes & Lye, 2024; Cranmer et al., 2020). These chiefly *impose* geometric structures via loss penalties, whereas MeshFT-Net *constrains* the architecture: the conservative interconnection is hard-wired by signed mesh incidences (topology), yielding a port–Hamiltonian form with learned metric/dissipation. Symplectic-ODE methods similarly preserve symplectic structure by integrating learned dynamics with symplectic schemes (Zhong et al., 2020), but they preserve a *nondegenerate* global symplectic two-form on finite-dimensional (often particle) phase spaces and do not expose the mesh chain–complex or separate *topology* from *metric*. By contrast, MeshFT-Net fixes the antisymmetric interconnection via incidence operators $D_k$ with $D_k D_{k+1} = 0$, enforcing topological identities that remove non-physical modes; learning is confined to metric maps. See Appendix C.3 for details.

Also, as another line of structure-preserving learning, metriplectic/GENERIC approaches preserve thermodynamic structure and typically adopt a pre-formulated global template whose operators are fitted with neural parameterizations (Zhang et al., 2022; Hernández et al., 2021; Lee et al., 2021; Hernández et al., 2025). In contrast, we deduce a local port–Hamiltonian reduction from minimal principles and use it as an architectural inductive bias without assuming a single global form a priori. We assume only the physical principles for the target dynamics. Our constraints come directly from principles, not from a pre-set model template. This is the key distinction we emphasize.

**Neural Constitutive Laws.** Neural constitutive laws (Ma et al., 2023) assume a continuous PDE and learns a constitutive law. In contrast, MeshFT does not assume continuum PDE: it works at the discrete level, fixes the conservative interconnection by mesh topology, and learns only the metric-dependent structure without PDE-residual supervision.

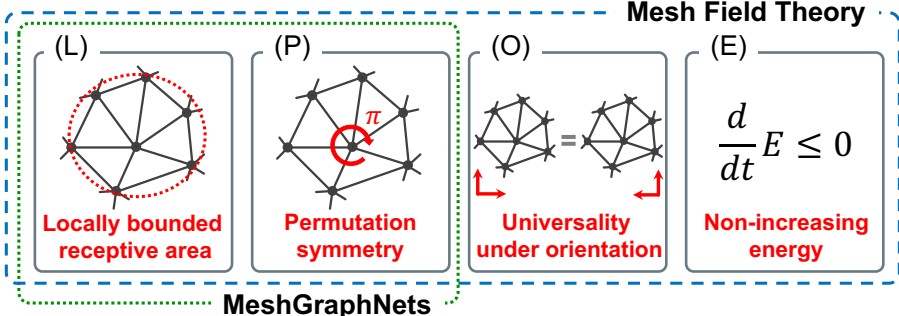

Figure 1: Core concept—comparison of MeshFT and MGN by underlying physical assumptions: (L) locality; (P) Permutation equivariance; (O) Orientation covariance; (E) Non-increasing energy. MGN attains (L) and (P) by architecture design, whereas MeshFT additionally enforces (O) and (E), yielding a clear modeling guideline: *fix topology* (incidence-based interconnection) and *learn metric-dependent structures*, which directly leads to MeshFT-Net.

**Discrete Exterior Calculus (DEC).** DEC gives a structure-preserving discrete counterpart of exterior calculus on meshes (Hirani, 2003; Desbrun et al., 2005a). Fields are $k$-cochains on $k$-cells; the discrete derivative is the coboundary (signed incidence matrix) $D_k$ with $D_{k+1}D_k = 0$ (discrete Stokes, $d^2=0$). Metric-dependent structure enters via the discrete Hodge star built from primal–dual volumes, inducing inner products and defining discrete energies. Constitutive laws use these metric-dependent operators. Thus DEC (and finite-element exterior calculus (Arnold et al., 2006)) makes the separation explicit: differential structure is topological, while metric-dependent structure appears only through constitutive relation. Beyond classical DEC, data-driven exterior calculus (DDEC) fixes exterior derivatives and learns Hodge operators from data (Trask et al., 2022).

**Exterior Calculus Related Graph Networks.** GNNs related to exterior calculus (e.g., GRAND (Chamberlain et al., 2021) and bracket-based GNN (Gruber et al., 2023).) cast layers as PDE/exterior-calculus flows for better interpretation and performance. While this work starts from a PDE-style template for network evolution, we start from minimal physical principles and derive a *local* port–Hamiltonian reduction on.

## 3 LOCAL REDUCTION OF MGN TO A PORT–HAMILTONIAN DYNAMICS

In this section, we establish a port–Hamiltonian formulation for mesh-based physics. To this end, we show that the dynamics of MGN reduce to a port–Hamiltonian form at the differential level under a minimal set of physical requirements, with several of these requirements already implicitly satisfied by vanilla MGN. Fig. 1 compares MeshFT and MGN in terms of their underlying assumptions, clarifying the paper's central idea. In particular, enforcing the orientation consistency and the energy balance eliminates non-physical degrees of freedom and reveals a clean separation between the topology-driven interconnection and the metric-dependent energetic part of the dynamics, thereby clarifying which must remain fixed and which components should be learned.

**Notation.** Let $\mathcal{K}$ be a finite oriented cell complex of dimension $d$. For each $k$, let $C^k \cong \mathbb{R}^{n_k}$ denote the space of real-valued $k$-cochains (one scalar per oriented $k$-cell), and let $D_k : C^k \to C^{k+1}$ be the signed incidence (coboundary) operator, with the usual property $D_{k+1}D_k = 0$ (Hirani, 2003; Desbrun et al., 2005a). We stack the cochain degrees that interact via incidence (e.g., $k$ and $k+1$) as $z := (z_k, z_{k+1}) \in C^k \oplus C^{k+1}$, and consider an autonomous update $\dot{z} = F(z)$. For example, a one-layer update in standard MGN often takes $z = (z_0, z_1)$ with node features $z_0 \in C^0$ and edge features $z_1 \in C^1$ on *undirected* edges. Hence, $z_1$ is not a DEC 1-cochain; the coupling is via permutation-invariant message passing rather than the signed incidence $D_0$. More generally, $z$ need not be limited to nodes and edges—it may include face features ($C^2$) or cell-centered features ($C^d$); vector-valued features are handled by replacing $C^k$ with $C^k \otimes \mathbb{R}^{r_k}$. Also, non-physical features (e.g., node/edge types) are excluded from the state $z$ here; they may only enter as fixed parameters of learnable maps, and are not treated as dynamical variables.

We equip the state $z = (z_k, z_{k+1}) \in C^k \oplus C^{k+1}$ with a storage function $H : C^k \oplus C^{k+1} \to \mathbb{R}_{\geq 0}$, strictly convex, and block-separable by degree, i.e., $H(z) = H_k(z_k) + H_{k+1}(z_{k+1})$. It is time-invariant but may encode spatial/material heterogeneity. Define the co-energy (conjugate) variable $e := \nabla H(z) \in C^k \oplus C^{k+1}$, where the gradient is taken with respect to the canonical Euclidean pairing, so the instantaneous power delivered to the state is $\langle e, \dot{z} \rangle = e^\top \dot{z}$. The Hessian $G(z) := \nabla^2 H(z) \succ 0$ is block-diagonal across degrees and acts as a state-dependent metric. We write the associated mass map as $M(z) := G(z)^{-1} \succ 0$. The linear–quadratic case is recovered when $G(z) \equiv M^{-1}$ is constant, i.e., $H(z) = \frac{1}{2} z^\top M^{-1} z$ and $e = M^{-1} z$.

## 3.1 What MGN Guarantees by Design, and What Physics Still Requires

In this section, we identify physical requirements that MGN implicitly respects and the essential ones absent from the standard formulation despite their importance in faithful physical modeling. In this section, we focus on the spatial structure and do not explicitly enforce additional temporal priors (e.g., causality), which are addressed by the causal time-stepping scheme.

**MGN Built-in Symmetry and Locality.** We note two inductive biases that MGN inherently satisfies—local interactions and permutation equivariance (label symmetry):

**Assumption 1** (MGN built-ins: Locality and Symmetry). *(L) Locality. The output on a $k$-cell may depend only on inputs within $L$ hops in the graph of $\mathcal{K}$. For $L = 1$, a $k$-cell receives only from itself, its own faces $(k - 1)$-cell, and its own cofaces $(k + 1)$-cell. (P) Permutation equivariance. For any permutation $\pi$ that preserves the partition of indices by degree/type, the predictor satisfies $f(\pi \cdot x) = \pi \cdot f(x)$; i.e., relabeling within each degree/type class only permutes the outputs.*

Informally, MGN realizes (L) and (P) by gathering messages (1-cochain) from 1-hop neighbor nodes (0-cell) with an order-independent reducer (e.g., sum/mean/max), applying a shared message/update kernel, and aggregating back to 1-hop neighbor nodes. This realizes permutation equivariance and spatial locality, which happens to match common physical desiderata—local and uniform governing laws. Intuitive visualization is shown in Fig. 1. However, only these assumptions do not guarantee geometric/orientation correctness or energetic consistency. With these in place, we formalize the two additional physical requirements not enforced by MGN—orientation covariance and energy balance.

**Physical Requirements Beyond MGN.** The following two physical requirements, typically not enforced by standard MGN, will be added:

**Assumption 2** (Orientation & Energy). *(O) Orientation covariance. Changing the sign convention of oriented entities (e.g., flipping edge/face directions) should only flip the signs of the corresponding oriented variables; scalar quantities such as energy and power must be unchanged. The formal flip-operator statement is given in Appendix A. (E) Energy balance and passivity. The dynamics split into a conservative part that never does net work and a dissipative part that never injects energy; consequently, in the absence of sources the stored energy cannot increase over time.*

Formally, we assume an energy balance with a conservative–dissipative split $F = F_{\mathrm{con}} + F_{\mathrm{diss}}$ and $e = \nabla H(z)$. *Pointwise* power satisfies, for all $z$, $e^\top F_{\mathrm{con}}(z) = 0$ and $e^\top F_{\mathrm{diss}}(z) \leq 0$, so $\dot{H} = e^\top \dot{z} = e^\top F(z) \leq 0$. We also impose the *incremental* (two-point) form in energy variables: for $z_1, z_2$ with $e_i = \nabla H(z_i)$, $(e_1 - e_2)^\top (F_{\mathrm{con}}(z_1) - F_{\mathrm{con}}(z_2)) = 0$, $(e_1 - e_2)^\top (F_{\mathrm{diss}}(z_1) - F_{\mathrm{diss}}(z_2)) \leq 0$. Thus the conservative part does no net work between states, while the dissipative part is *monotone* in $e$. If $F$ is differentiable, these incremental conditions are equivalent to $\mathrm{Sym}\big(\frac{\partial F_{\mathrm{con}}}{\partial e}\big) = 0$ and $\mathrm{Sym}\big(\frac{\partial F_{\mathrm{diss}}}{\partial e}\big) \preceq 0$ which yields the skew/dissipative split used in our reduction.

**Importance of Orientation Covariance** Orientation is a sign gauge: flipping the orientation of $k$-cells (edge arrows, face normals) changes coordinates but not physics. Let $\rho = \mathrm{diag}(\rho_0, \ldots, \rho_d)$ with $\rho_k \in \{\pm I\}$ act on all oriented variables on $k$-cells. Scalars are gauge-invariant: $H(\rho z) = H(z)$ and $e(\rho z)^\top F(\rho z) = e(z)^\top F(z)$. Fluxes carried by oriented $k$-cells are gauge-covariant: $q_k \mapsto \rho_k q_k$. The signed incidence transforms as $D_k \rho_k = -D_k$, $\rho_{k+1} D_k = -D_k$, and $\rho_{k+1} D_k \rho_k = D_k$, corresponding to flipping degree $k$ only, degree $k+1$ only, or both simultaneously. Assumption 2 (O) ensures sign-gauge equivariance: the physical laws retain their form and scalar pairings (e.g., $e^\top \dot{z}$) remain unchanged under orientation flips. In other words, this guarantees that physical system is universal regardless of the choice of mesh orientation shown in Fig. 1.

## 3.2 Theorem: Local Reduction to Port–Hamiltonian Dynamics

We show that mesh-based dynamics of MGN satisfying the built-in biases (locality and permutation equivariance) together with the physical principles introduced above—orientation covariance and energy balance/passivity—admit a *local* reduction to port–Hamiltonian representation.

We now state the reduction; a complete proof of Theorem 1 appears in Appendix A.

**Theorem 1** (Local reduction to port–Hamiltonian dynamics). *Let $F : C^k \oplus C^{k+1} \to C^k \oplus C^{k+1}$ be a mesh network defining the continuous-time dynamics $\dot{z} = F(z)$. Assume (L), (P), (O), and (E). Then, at any point where the Jacobian exists, there is a local energy reparameterization under which*

$$\frac{\partial F}{\partial z}(z) = \big(J - R(z)\big)\, G(z), \qquad J^\top = -J, \;\; R(z) \succeq 0, \;\; G(z) \succ 0, \tag{1}$$

*where $J$ is assembled from the signed incidence matrices $\{D_k\}$ and thus depends only on topology.*

**Remark.** Importantly this theorem is a equation-agnostic local reduction. Thus, we do not assume or claim a single global port–Hamiltonian form for the underlying dynamics. This theorem claims that when a (possibly unknown) $F(z)$ is locally differentiable and satisfies the principles, its Jacobian admits the factorization. Conversely, if $F$ is only piecewise smooth (e.g., has discontinuities), the Jacobian may fail to exist on the interface, that is, the reduction holds only piecewise.

Here, we focus on the simplest case of a *state-independent* interconnection $J$. When state-dependent couplings are needed, we retain the same incidence-based skew sparsity and modulate each block by local orientation-even gains, i.e., $J \to J(z)$; the local reduction still holds pointwise; see Appendix B. When $J$ depends only on geometry/material (i.e., not on the current state z), any geometry/material-dependece factors can be absorbed into the energy metric $G$, yielding a constant, purely incidence-based (topological) wiring; see the remark in Appendix A.4

*Limitation and Practical Note.* If one allows state-dependent interconnection, the same incidence sparsity and blockwise skew structure hold *at each state $z$*, but no scaling factors can be absorbed into $G$ and $J$ need not be constant; see the remark in Appendix A.4. In continuum physics (e.g., ideal fluids and ideal magnetohydrodynamics (MHD)), energy-preserving couplings often depend on the fields; this arises from the Poisson-bracket structure in such systems (Morrison & Greene, 1980; Morrison, 1982). Practically, because the sparsity pattern is fixed by topology, learning reduces to estimating the (possibly state-/geometry-/material-dependent) coefficients that scale the interconnection blocks of $J$, under (P) permutation equivariance and (O) orientation covariance. The incidence wiring $D_k$ is known a priori and need not be learned. To clarify the structure and strengths of MeshFT and MeshFT-Net, the main text focuses on the state-independent case—which already covers all canonical Hamiltonian settings (e.g., linear waves and linear electromagnetics/elasticity); the state-dependent extension is straightforward and appears in Appendix B.

As a consequence, we may write $F(z) = J\nabla H(z) + F_{\text{diss}}(z)$, with $\frac{\partial F_{\text{diss}}}{\partial e}(z) = -R(z) \succeq 0$. If dissipation is absent, set $F_{\text{diss}} \equiv 0$ (equivalently $R \equiv 0$). Then $\dot{z} = Je$, $\dot{H} = e^\top F = e^\top Je = 0$, so the flow conserves energy; in the general case, $\dot{H} = e^\top F = e^\top F_{\text{diss}} \le 0$. In addition, while beyond the present scope, external source/supply terms can be also introduced in the standard port–Hamiltonian manner without altering the topology-determined interconnection $J$.

The novelty here is not merely to assert a port–Hamiltonian form, but to prove what is fixed and learnable and to identify sufficient conditions for this separation. Under (L), (P), (O), and (E)—independently of model architecture—the conservative interconnection $J$ is *uniquely determined* by mesh topology, while geometry/material and dissipation enter only through $G$ and $R$. Hence learning reduces to estimating $G$ and $R$, with $J$ kept fixed.

## 4 MeshFT-Net: Neural Realization of MeshFT

We now instantiate the reduction as an architecture. We can consider two instantiations consistent with the differential form above: **General.** Parameterize a strictly convex storage $H_\theta$ (and a convex dissipation potential $\Psi_\theta$). Here $\Psi_\theta$ is a convex function whose gradient yields the dissipative force in the energy variables. With the co-energy $e = \nabla H_\theta(z)$, define the dynamics in energy coordinates by $\dot{z} = Je - \nabla_e \Psi_\theta(e)$, so that $\partial F/\partial e = J - \nabla_e^2 \Psi_\theta(e)$ and $G(z) = \nabla^2 H_\theta(z)$. This covers

---

**Algorithm 1** MeshFT-Net: One-Layer Update

---

**Require:** step $\Delta t$; fixed $J = \begin{pmatrix} 0 & -D_k^\top \\ D_k & 0 \end{pmatrix}$; learnable SPD $G_\theta$; PSD $R_\theta$; inputs $z^n = (z_k^n, z_{k+1}^n)$

**Ensure:** outputs $z^{n+1} = (z_k^{n+1}, z_{k+1}^{n+1})$)

    **(A) Half-damp (in)**

1: $z_k^{n,-} \leftarrow \exp\left( -\frac{\Delta t}{2} R_{k,\theta}\big(\{z_k^n, z_{k+1}^n\}\big) G_k \right) z_k^n$

2: $z_{k+1}^{n,-} \leftarrow \exp\left( -\frac{\Delta t}{2} R_{k+1,\theta}\big(\{z_k^n, z_{k+1}^n\}\big) G_{k+1} \right) z_{k+1}^n$      *note:* $\exp(\cdot)$ *is the matrix exponential.*

    **(B) Conservative pass (KDK under $J$)**

3: $z_k^{\text{half}} \leftarrow z_k^{n,-} - \frac{\Delta t}{2} D_k^\top\big(G_{k+1} z_{k+1}^{n,-}\big)$

4: $z_{k+1}^{\text{pre}} \leftarrow z_{k+1}^{n,-} + \Delta t\, D_k\big(G_k z_k^{\text{half}}\big)$

5: $z_k^{\text{pre}} \leftarrow z_k^{\text{half}} - \frac{\Delta t}{2} D_k^\top\big(G_{k+1} z_{k+1}^{\text{pre}}\big)$

6: $z^{\text{pre}} \leftarrow \{z_k^{\text{pre}}, z_{k+1}^{\text{pre}}\}$   (CFLGUARD($\Delta t$))

    **(C) Half-damp (out)**

7: $z_k^{n+1} \leftarrow \exp\left( -\frac{\Delta t}{2} R_{k,\theta}(z^{\text{pre}}) G_k \right) z_k^{\text{pre}}$

8: $z_{k+1}^{n+1} \leftarrow \exp\left( -\frac{\Delta t}{2} R_{k+1,\theta}(z^{\text{pre}}) G_{k+1} \right) z_{k+1}^{\text{pre}}$

---

nonlinear constitutive laws. **Quadratic first-order model.** For efficiency we use a quadratic storage and a first-order (Picard) linearization around the current state. With $H_\theta(z) = \frac{1}{2} z^\top G_\theta z$ (degreewise $G_\theta \succ 0$, state-independent), we have $e = G_\theta z$ and $\dot{z} \approx \big(J - R_\theta(z)\big) e = \big(J - R_\theta(z)\big) G_\theta z$. In experiments presented in this paper, we adopt this quadratic, state-independent $G_\theta$.

**Fixed vs. Learned.** By Theorem 1, the conservative interconnection is incidence-only, $J = \begin{pmatrix} 0 & -D_k^\top \\ D_k & 0 \end{pmatrix}$, so we *do not train $J$*. The *learnable* components are $G_\theta$ (degreewise symmetric positive-definite, SPD; shared within each degree/type and encoding geometry/material) and $R_\theta$ (optional dissipation; positive-semidefinite, PSD).

**Time Stepping (one-layer update).** The integrator is one design choice rather than unique, other numerically consistent variants can be also used. Here, we advance the state with a Strang splitting (Strang, 1968); one concrete realization is given in Algorithm 1. CFLGUARD($\Delta t$) in Algorithm 1 scales the step (or selects substeps) to satisfy a target CFL condition (Courant et al., 1967; LeVeque, 1992). Also, all heavy operations here are sparse matrix–vector products, yielding $O(N)$ time and memory, where N is the total number of degrees of freedom.

**Parameterization.** $G_\theta$ is degreewise SPD (positivity by construction), implemented as diagonals (softplus) or small Cholesky blocks, optionally conditioned by permutation-equivariant, orientation-even local MLPs using geometry/material features. $R_\theta(z)$ is PSD (e.g., Rayleigh-type $z \mapsto \gamma(\cdot) G_\theta^{-1} z$); state dependence enters only through $R_\theta(z)$.

**Training.** Given $z^n$, compute the layer output $\hat{z}^{n+1} = \text{MeshFT-Net}_{\Delta t}\big(z^n; J, G_\theta, R_\theta\big)$. Training uses a supervised one–step loss, e.g. $\sum_{k \in \mathcal{I}} \text{Loss}\big(\hat{z}_k^{n+1}, z_k^{n+1}\big)$, where $\mathcal{I}$ may be all components or a chosen subset. Optionally, this time-stepping can also be *stacked*, with supervision applied only to the final output composed of sub-step evolution. No PDE–residual terms are used; the inductive bias comes from the fixed interconnection $J$ and the SPD/PSD structure of $G_\theta$ and $R_\theta$.

## 5 EXPERIMENTS

We test the key implication of Theorem 1: after eliminating non-physical degrees of freedom, the predictor has a port–Hamiltonian form where the interconnection $J$ is fixed by mesh topology, and learning targets only the metric $G$ and dissipation $R$. We hypothesize that this structural prior preserves long-horizon stability, increases physical fidelity, and improves data efficiency. For comparisons with HNN, we work in canonical variables $(x_k, p_k)$, where $p_k$ is the momentum conjugate to $x^{(k)}$, rather than using flux variables $x_{k+1}$; see Appendix D for details.

**Baselines.** Consider four graph-based simulators, ranging from unconstrained to structure-preserving. All models are trained on the same data underidentical training protocols and share

Table 1: One-step MSE, normalized energy drift ($\Delta E/E_0$), and TSMSE over the rollout horizon on regular grids. Lower is better; drift closer to $0$ is better. All numbers are computed on held-out validation data. Mean $\pm$ s.d. over $5$ seeds are reported.

| Model | One-step MSE | TSMSE | Energy drift $\Delta E/E_0$ |
|---|---|---|---|
| MeshFT-Net | $\mathbf{1.3 \times 10^{-9} \pm 5.6 \times 10^{-10}}$ | $\mathbf{9.6 \times 10^{-5} \pm 2.6 \times 10^{-5}}$ | $\mathbf{1.3 \times 10^{-4} \pm 2.1 \times 10^{-5}}$ |
| MGN | $1.6 \times 10^{-7} \pm 6.1 \times 10^{-8}$ | $1.3 \times 10^{-1} \pm 9.8 \times 10^{-2}$ | $25.9 \pm 40.8$ |
| MGN-HP | $5.7 \times 10^{-4} \pm 1.8 \times 10^{-5}$ | $6.1 \times 10^{-1} \pm 1.0 \times 10^{-1}$ | $16.0 \pm 6.8$ |
| HNN | $3.5 \times 10^{-8} \pm 5.2 \times 10^{-8}$ | $3.0 \times 10^{-3} \pm 3.2 \times 10^{-3}$ | $1.0 \times 10^{-2} \pm 1.5 \times 10^{-2}$ |

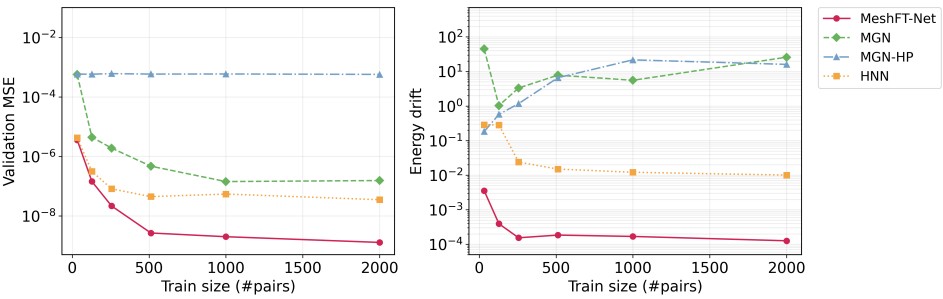

Figure 2: Relationship between the size of training dataset and one-step MSE (left) and rollout energy drift (right) for regular grid mesh.

same symmetric second-order integrator. To isolate architectural differences, we deliberately chose pedagogically clear, structurally comparable baselines.

**Ours: MeshFT-Net.** A structure-preserving model based on Theorem 1: the interconnection is fixed by signed incidences, while the metric $G_\theta$ and optional dissipation $R_\theta$ are learned. **MGN.** A structure-agnostic message-passing network that predicts $\dot{z} = v_\phi(z)$ from node/edge features without enforcing physical structure. **MGN-HP (MGN with Hamiltonian Penalty).** An MGN augmented with a learned scalar energy $H_\theta(z)$ and a penalty that aligns $v_\phi(z)$ with the Hamiltonian vector field $X_{H_\theta}(z)$ (in canonical coordinates $X_{H_\theta}(q,p) = (\partial_p H_\theta, -\partial_q H_\theta)$). This preserves MGN's flexibility while nudging it toward conservative dynamics. **HNN.** A Hamiltonian model that learns $H_\theta(z)$ and defines the field by $X_{H_\theta}$; training minimizes derivative mismatch $\|\dot{z} - X_{H_\theta}(z)\|^2$. The Hamiltonian structure is *embedded in the loss*. We instantiate the common separable form $H(q,p) = U(q) + T(p)$, though nonseparable $H_\theta$ is also compatible.

### 5.1 Analytic Plane-Wave Benchmark

We evaluate each model with periodic 2D plane waves on regular grids and Delaunay triangulations. Metrics are (i) one-step mean squared error (MSE), (ii) time-series mean squared error (TSMSE) that is the average MSE over the full rollout horizon, and (iii) normalized energy drift over open-loop rollouts. We also vary the training set size. On regular grids (Table 1), MeshFT-Net attains the lowest error and drift with near-zero power, while MGN, MGN-HP, and HNN show orders-of-magnitude larger drift and spurious power injection. The same ordering holds on Delaunay meshes. Across data sizes (Figs. 2, 4), MeshFT-Net maintains near-zero drift and achieves about $5$ times data efficiency vs. MGN. HNN enforces Hamiltonian structure with soft penalties. It improves stability over vanilla MGN, MeshFT-Net's topological interconnection with a symplectic step yields orders of magnitude greater robustness. Implementation details and additional results appear in Appendix D.1.

We also test a Rayleigh-damped setting (amplitude $\propto e^{-\gamma t}$); details appear in Appendix D.2. For HNN, we also learn an explicit Rayleigh damping term same as MeshFT-Net. We report one-step MSE, the time-series mean squared error (TSMSE) over the full rollout horizon, and the *normalized energy error* (NEE) relative to the theoretical energy. As shown in Table 2, MGN attains the lowest one-step error ($5.2 \times 10^{-8} \pm 2.2 \times 10^{-8}$), while MeshFT-Net achieves the best energy fidelity (NEE $2.1 \times 10^{-2} \pm 3.8 \times 10^{-3}$ and the lowest TSMSE ($2.1 \times 10^{-2} \pm 2.0 \times 10^{-3}$). However, relative to

Table 2: Dissipative benchmark. One-step MSE is the per-step prediction error. TSMSE is the time-series mean squared error over the full rollout horizon. NEE is the normalized energy error relative to the theoretical energy. All metrics are lower is better. All numbers are computed on held-out validation data. Mean $\pm$ s.d. over 3 seeds are reported.

| Model | One-step MSE | TSMSE | NEE |
|---|---|---|---|
| MeshFT-Net | $1.2 \times 10^{-7} \pm 3.4 \times 10^{-9}$ | $\mathbf{2.1 \times 10^{-2} \pm 2.0 \times 10^{-3}}$ | $\mathbf{2.1 \times 10^{-2} \pm 3.8 \times 10^{-3}}$ |
| MGN | $\mathbf{5.2 \times 10^{-8} \pm 2.2 \times 10^{-8}}$ | $1.7 \times 10^{-1} \pm 4.4 \times 10^{-2}$ | $2.2 \pm 1.0$ |
| MGN-HP | $4.5 \times 10^{-4} \pm 1.3 \times 10^{-5}$ | $1.9 \pm 5.0 \times 10^{-1}$ | $5.0 \pm 1.9$ |
| HNN | $2.5 \times 10^{-7} \pm 3.8 \times 10^{-7}$ | $2.2 \times 10^{-1} \pm 7.5 \times 10^{-2}$ | $0.49 \pm 0.16$ |

Table 3: Physical-consistency and learning-adequacy diagnostics. Physical-consistency uses $T{=}200$, $\Delta t{=}0.002$ (lower is better). PDE residuals are reported as short/long; the short residual is computed over $T{=}5$ and the long residual over $T{=}200$. Learning-adequacy uses $T_{\text{short}}{=}16$ (higher is better for cosine; lower otherwise). All numbers are computed on held-out validation data. Mean over 5 seeds is reported. Best values are shown in bold, and second-best values are underlined.

(a) Physical-consistency

| Model | Wave-speed | Canonical | PDE resid. short/long | Equipartition | Momentum |
|---|---|---|---|---|---|
| MeshFT-Net | $\mathbf{2.03 \times 10^{-2}}$ | $\mathbf{9.63 \times 10^{-6}}$ | $\mathbf{3.61 \times 10^{-3}/3.32 \times 10^{-3}}$ | $\mathbf{4.11 \times 10^{-2}}$ | $\mathbf{4.89 \times 10^{-8}}$ |
| MGN | $2.00 \times 10^{-1}$ | $1.02 \times 10^{-3}$ | $3.56 \times 10^{-2}/2.68 \times 10^{-1}$ | $\underline{1.68 \times 10^{-1}}$ | $3.90 \times 10^{-1}$ |
| MGN-HP | $3.91 \times 10^{-1}$ | $1.65 \times 10^{-3}$ | $1.11 \times 10^{-1}/2.21 \times 10^{-1}$ | $2.60 \times 10^{-1}$ | $\underline{9.01 \times 10^{-2}}$ |
| PI-MGN | $4.84 \times 10^{-1}$ | $15.9$ | $2.48 \times 10^{-2}/6.68 \times 10^{-1}$ | $4.39 \times 10^{-1}$ | $4.99$ |
| HNN | $8.88 \times 10^{-2}$ | $\underline{6.49 \times 10^{-5}}$ | $1.88 \times 10^{-2}/\underline{2.01 \times 10^{-1}}$ | $2.01 \times 10^{-1}$ | $1.07$ |
| FNO | $\underline{3.08 \times 10^{-1}}$ | $6.95 \times 10^{-2}$ | $\underline{5.09 \times 10^{-2}}/5.81 \times 10^{-1}$ | $2.60 \times 10^{-1}$ | $1.65$ |
| GraphCON | $2.11 \times 10^{-1}$ | $2.42 \times 10^{-2}$ | $1.57 \times 10^{-1}/2.80 \times 10^{-1}$ | $1.79 \times 10^{-1}$ | $3.00 \times 10^{-1}$ |

(b) Learning-adequacy

| Model | VF cosine ($\uparrow$) | VF $L_2$ | Short roll MSE | Amp err | Phase err (deg) |
|---|---|---|---|---|---|
| MeshFT-Net | $\mathbf{0.999996}$ | $\mathbf{3.03 \times 10^{-3}}$ | $\underline{9.36 \times 10^{-2}}$ | $\mathbf{1.25 \times 10^{-2}}$ | $\mathbf{8.06 \times 10^{-1}}$ |
| MGN | $0.999882$ | $2.85 \times 10^{-2}$ | $\underline{1.06 \times 10^{-1}}$ | $5.05 \times 10^{-2}$ | $2.47$ |
| MGN-HP | $0.999353$ | $2.80 \times 10^{-2}$ | $2.11 \times 10^{-1}$ | $1.66 \times 10^{-1}$ | $5.99$ |
| PI-MGN | $0.935556$ | $2.96 \times 10^{-1}$ | $2.98 \times 10^{-1}$ | $6.93 \times 10^{-2}$ | $13.9$ |
| HNN | $\underline{0.999922}$ | $\underline{1.14 \times 10^{-2}}$ | $\mathbf{6.34 \times 10^{-2}}$ | $\underline{2.14 \times 10^{-2}}$ | $\underline{1.05}$ |
| FNO | $0.999821$ | $\underline{2.32 \times 10^{-2}}$ | $1.26 \times 10^{-1}$ | $\underline{8.29 \times 10^{-2}}$ | $3.98$ |
| GraphCON | $0.998779$ | $4.62 \times 10^{-2}$ | $1.96 \times 10^{-1}$ | $1.26 \times 10^{-1}$ | $4.60$ |

MeshFT-Net, MGN is $1.0 \times 10^{2}$ times worse in NEE. These results indicate that fixing the incidence-based interconnection and learning only metric/dissipation captures dissipation most faithfully.

## 5.2 Physics-Consistency Benchmark

Beyond accuracy and energy drift, we ask whether models respect the *physics* on dissipation-free 2D plane waves (periodic grids). We use model-agnostic diagnostics shared by all baselines: (i) wave speed error, (ii) canonical consistency, (iii) PDE residual, (iv) kinetic–potential equipartition, and (v) momentum conservation. We also run lightweight learning-validity checks: vector-field alignment and amplitude/phase fit. Full definitions appear in Appendix D.3. To broaden coverage, we additionally include a neural operator (FNO) (Li et al., 2021), a long-range graph simulator (GraphCON) (Rusch et al., 2022), and a physics-informed MeshGraphNet (PI-MGN) (Würth et al., 2024) alongside MGN, MGN-HP, and HNN. Across all five diagnostics (Table 3 (a)), MeshFT-Net is most physically faithful: smallest wave-speed error ($\sim 10^{-2}$), near-exact canonical relation ($\sim 10^{-6}$), minimal PDE residual ($\sim 10^{-3}$), closest equipartition, and essentially zero momentum change ($\sim 10^{-8}$), outperforming the baselines on all metrics, often by orders of magnitude. Momentum

Table 4: OOD generalization under three shifts of Frequency, Parameter, and Resolution (lower is better for all metrics). Columns report TSMSE and normalized energy drift, computed on held-out validation sets. Values are means over 3seeds. Bold indicates the best and underline the second best in each column. Entries shown as $> 100$ exceeded the evaluation cap (diverged). Full results are in Appendix D.4.

| Model | Frequency | | Parameter | | Resolution | |
|---|---|---|---|---|---|---|
| | TSMSE | Drift | TSMSE | Drift | TSMSE | Drift |
| MeshFT-Net | **0.18** | $5.1{\times}10^{-3}$ | 0.71 | **0.17** | $5.1{\times}10^{-2}$ | $2.9{\times}10^{-3}$ |
| MGN | 9.0 | $>100$ | 1.7 | 25.5 | 0.53 | 3.4 |
| MGN-HP | 0.62 | 0.16 | **0.59** | 0.25 | 0.58 | 5.6 |
| PI-MGN | $>100$ | $>100$ | $>100$ | $>100$ | $>100$ | $>100$ |
| HNN | 1.54 | 0.45 | 0.80 | 0.24 | 0.36 | 2.0 |
| FNO | 1.94 | 2.7 | 0.96 | 1.6 | $>100$ | $>100$ |
| GraphCON | 69.0 | 96.4 | $>100$ | $>100$ | 0.32 | 0.71 |

is preserved without explicit constraints because MeshFT-Net enforces action–reaction at interfaces guaranteed by (O); the other baselines lacking this property need not conserve it. The added baselines follow the same trend: FNO and GraphCON achieve reasonable short-horizon accuracy yet exhibit larger long-horizon residuals and momentum drift, while PI-MGN improves some accuracy terms but degrades canonical consistency and long-horizon residuals. Learning-validity checks (Table 3 (b)) reinforce the advantage of MeshFT-Net: it achieves the best alignment, second-smallest short-horizon error, and the most accurate amplitude/phase recovery. While a soft Hamiltonian penalty (as in HNN) can slightly improve physical consistency, MeshFT-Net achieves substantially more robust physical consistency. Overall, fixing topology-driven interconnection while learning metric/dissipation yields physically consistent predictions.

## 5.3 OOD Generalization

We also evaluate OOD generalization on periodic 2D waves under three shifts—(i) frequency ($k_{\max}^{\text{test}} > k_{\max}^{\text{train}}$), (ii) resolution (coarse→fine), and (iii) parameter ($c_{\text{test}} \neq c_{\text{train}}$). Metrics are TSMSE and normalized energy drift after a fixed-horizon rollout; details and full results in Table 8 of Appendix D.4. This tests whether inductive bias—not just capacity—supports accurate, physically consistent rollouts under spectral, discretization, and parameter shifts. We include baselines MGN, MGN-HP, PI-MGN, HNN, FNO, and GraphCON. Across all OOD shifts in Table 4, MeshFT-Net delivers the lowest energy drift overall and the best TSMSE on *Frequency* and *Resolution*. Several baselines diverge under certain shifts ($> 100$), whereas MeshFT-Net remains stable and accurate across spectral, discretization, and parameter changes. Importantly, together with the one-step MSE shown in Table 8, these results show that strong local generalization does not necessarily translate into faithful long-horizon dynamics or energy stability. Models with competitive one-step MSE can still exhibit larger TSMSE and/or drift over rollouts. Overall, fixing the incidence-based interconnection while learning only metric/dissipation extrapolates robustly across spectral, resolution, and parameter shifts, whereas unconstrained MGN degrades in both error and drift out of distribution.

## 5.4 Validation on Acoustic Scattering Benchmark a from *The Well*

To assess transfer beyond synthetic data, we evaluate MeshFT-Net on a subset of *The Well*—Acoustic Scattering (Discontinuous) (Ohana et al., 2024) (see also (Mandli et al., 2016))—which is near-Hamiltonian but includes discontinuous media and open/reflective boundaries. We train with one-step teacher forcing for the pressure field. This probes whether a topology-fixed interconnection that preserves structure also maintains fidelity for more realistic data. The details of this experiment appear in Appendix D.5. Fig. 3 shows the snapshots of predicted and ground-truth pressure fields on the validation set, using a common colormap. MeshFT-Net closely matches wavefront position, curvature, and interference nodes; differences are limited to slight smoothing of high-frequency details and reduced peak contrast, while boundary reflections remain consistent. These visuals agree with the quantitative findings of low drift and correct dispersion.

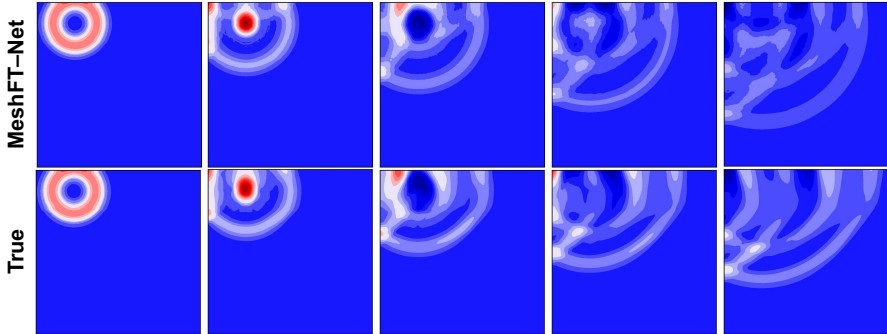

Figure 3: Pressure snapshots at equal time steps ordered right to left; the rightmost frame is the initial state (shared colormap).

Table 5: Results on nonlinear advection–diffusion under varying time step $\Delta t$ and training set size.

| $\Delta t$ | **Training Data** | **TSMSE** | **Mass Drift** |
|---|---|---|---|
| $5.0{\times}10^{-5}$ | 400 | $5.1{\times}10^{-6} \pm 6.7{\times}10^{-6}$ | $5.4{\times}10^{-4} \pm 7.6{\times}10^{-4}$ |
| $1.0{\times}10^{-3}$ | 400 | $7.4{\times}10^{-5} \pm 4.7{\times}10^{-5}$ | $2.8{\times}10^{-2} \pm 4.0{\times}10^{-2}$ |
| $5.0{\times}10^{-5}$ | 4000 | $4.6{\times}10^{-6} \pm 6.0{\times}10^{-6}$ | $4.4{\times}10^{-4} \pm 6.1{\times}10^{-4}$ |
| $1.0{\times}10^{-3}$ | 4000 | $1.3{\times}10^{-4} \pm 4.2{\times}10^{-5}$ | $1.3{\times}10^{-5} \pm 3.6{\times}10^{-6}$ |

### 5.5 Discussion on Nonlinear Advection–Diffusion

To assess generality, we evaluate MeshFT-Net on a weakly nonlinear advection–diffusion dataset and vary both the training set size and the physical time step $\Delta t$ used to generate input–output pairs. Averaged over three seeds (Table 5), the model attains low trajectory error (TSMSE). TSMSE increases at larger $\Delta t$, as expected from sparser supervision induced by coarser time resolution, but the degradation is modest and does not indicate qualitative breakdown. Changing the training size has only a secondary effect compared to $\Delta t$. Mass drift is generally small, but becomes noticeable in the large $\Delta t$, low-data regime, a sparse-supervision setting in time. This instability is mitigated when sufficient training data are available. Full experimental details appear in Appendix D.6. These findings indicate that, beyond architectural inductive bias, the interplay between data sparsity (via $\Delta t$) and data budget affects long-horizon stability, which is not captured by TSMSE alone.

## 6 Conclusion

We established *Mesh Field Theory (MeshFT)*: under four physical principles—locality, permutation equivariance, orientation covariance, and energy balance/dissipation inequality—mesh-based physics locally reduce to a port–Hamiltonian form in which the conservative interconnection is *uniquely determined* by mesh topology, while metric effects enter only through constitutive relations and dissipation. The contribution is not merely to assert the existence of a port–Hamiltonian structure, but to *prove* which components are structurally fixed and learnable and to identify sufficient physical conditions for this identification, thereby unifying ideas from DEC with learnable models.

Building on the reduction theorem, we designed MeshFT-Net. Across analytic datasets and real acoustic data, physics-consistency tests, and OOD validation, MeshFT-Net delivers accurate, long-horizon-stable, and physically consistent rollouts, achieves higher data efficiency over baselines, and exhibits robust extrapolation across frequency, resolution, and parameter shifts. These gains follow from the modeling principle given by the reduction: the *topological structure has no learnable freedom*—the incidence-based interconnection is predetermined—while *learning is confined to the metric* (constitutive operators and dissipation). This topology/metric separation removes non-physical degrees of freedom, yields interpretable parameters, and provides a rigorous inductive bias for stable, faithful, and data-efficient simulation. Taken together, these findings lay the foundation for a *MeshFT*: topology determines the interconnection, and the metric is what we need to learn.

## A    PROOF AND TECHNICAL DETAILS TO THEOREM 1

We adopt the notation of the main text. The state is $z$, the storage is a strictly convex energy $H$, the co-energy is $e = \nabla H(z)$, and the Hessian $G(z) = \nabla^2 H(z) \succ 0$. The dynamics split as $\dot{z} = F(z) = F_{\mathrm{con}}(z) + F_{\mathrm{diss}}(z)$ with conservative power balance $e^\top F_{\mathrm{con}}(z) = 0$ and passivity $e^\top F_{\mathrm{diss}}(z) \leq 0$ (no sources).

**Orientation/Sign Conventions.**    Let $\mathcal{K}$ be an oriented cell complex with cochain spaces $\{C^k\}_{k=0}^d$ and signed coboundaries $D_k : C^k \to C^{k+1}$ satisfying $D_{k+1}D_k = 0$. Degree-wise flips are encoded by $\rho = \mathrm{diag}(\rho_0, \ldots, \rho_d)$, where each $\rho_k$ is diagonal with $\pm 1$ entries (orientation gauge). Consistent flips leave incidences invariant, $\rho_{k+1}D_k\rho_k = D_k$, whereas single-degree flips change sign: $D_k\rho_k = -D_k$ and $\rho_{k+1}D_k = -D_k$. Orientation covariance (O) is expressed by

$$H(\rho z) = H(z), \qquad \nabla H(\rho z) = \rho \nabla H(z), \qquad F(\rho z) = \rho F(z), \tag{2}$$

so scalar quantities (energy, power) are gauge-invariant while flux-like quantities co-transform with their carriers. These conventions will be used repeatedly in the proofs below.

### A.1    LOCAL SYMMETRY BASIS AND ORIENTATION COVARIANCE

We first formalize the structure forced by (L), (P), and orientation covariance (O).

**Lemma 1** (Local, permutation-equivariant linear basis)**.** *Let $T : C^{k-1} \times C^k \times C^{k+1} \to C^k$ be linear, interface-local (depends only on cells incident to the output $k$-cell), and permutation-equivariant (with permutations acting within each degree/type class). Then $T$ decomposes as*

$$T(x_{k-1}, x_k, x_{k+1}) = a\, x_k\, +\, b\, A_k x_k\, +\, \alpha\, D_{k-1}x_{k-1}\, -\, \beta\, D_k^\top x_{k+1}, \tag{3}$$

*where $A_k$ is the* unsigned *adjacency of $k$-cells on $\mathcal{K}$, and coefficients $(a, b, \alpha, \beta)$ are label-independent scalars.*

*Proof.* (L) implies that the output on any $k$-cell can only use: itself, adjacent $k$-cells (sharing an interface), its $(k{-}1)$ faces, and its $(k{+}1)$ cofaces. From (P), relabeling cells within a degree/type must only relabel the output, so values within the same relation type (self, $k$-neighbors, faces, cofaces) must share one common weight. With linearity, the only degree-compatible linear maps supported on this incident neighborhood are the identity on $C^k$, the unsigned $k$–$k$ adjacency $A_k : C^k \to C^k$, the boundary $D_{k-1} : C^{k-1} \to C^k$, and the coboundary $-D_k^\top : C^{k+1} \to C^k$.

Hence $T$ is the stated linear combination. Any other term would either use non-incident cells (violating locality) or assign different weights within a relation type (violating permutation equivariance). $\qquad\square$

**Lemma 2** (Orientation covariance rules out same-degree terms)**.** *From* (O)*, in the decomposition of Lemma 1, we must have $a = b = 0$. Consequently,*

$$T(x_{k-1}, x_k, x_{k+1}) \;=\; \alpha\, D_{k-1}x_{k-1}\, -\, \beta\, D_k^\top x_{k+1}. \tag{4}$$

*Proof.* By (O), if we change the sign convention only at degree $k$, then the $k$-oriented output flips sign accordingly:

$$\rho_k^{-1}\, T(x_{k-1},\, \rho_k x_k,\, x_{k+1}) \;=\; -\, T(x_{k-1}, x_k, x_{k+1}) \quad \text{for all inputs.}$$

Evaluate the basis terms from Lemma 1 under this operation. Since $A_k$ is unsigned and depends only on incidence, it commutes with $\rho_k$. Hence

$$\rho_k^{-1}(a\, \rho_k x_k) = a\, x_k, \qquad \rho_k^{-1}(b\, A_k\, \rho_k x_k) = b\, A_k x_k,$$

while the cross-degree terms change sign on the $k$-side:

$$\rho_k^{-1}\big(D_{k-1}x_{k-1}\big) = -\, D_{k-1}x_{k-1}, \qquad \rho_k^{-1}\big(D_k^\top x_{k+1}\big) = -\, D_k^\top x_{k+1}.$$

Therefore the left-hand side equals $a\, x_k + b\, A_k x_k - \alpha\, D_{k-1}x_{k-1} + \beta\, D_k^\top x_{k+1}$. Since this must be the negative of $T(x_{k-1}, x_k, x_{k+1})$ for all inputs, the within-degree part must vanish, i.e., $a = b = 0$.

Finally, (O) also requires consistency under simultaneous flips across degrees. Using the standard sign behavior,

$$\rho_k^{-1}D_{k-1}\rho_{k-1} = D_{k-1}, \qquad \rho_k^{-1}D_k^\top \rho_{k+1} = D_k^\top,$$

so the remaining cross-degree terms satisfy (O). This yields the stated form. $\qquad\square$

**Note on Same-Degree Adjacency.** If we split $A_k$ into lower/upper $k$–$k$ adjacencies (sharing a $(k{-}1)$- or a $(k{+}1)$-cell with independent weights, (O) still forces both weights to be zero. Indeed, both adjacency operators commute with the degree-$k$ flip, while (O) requires $T$ to change sign under that flip (after compensating the output sign). Hence Lemma 2 and its consequences remain unchanged.

A.2   SKEW/DISSIPATIVE SPLIT FROM ENERGY BALANCE

The next lemma formalizes, at the differential level, the split implied by the incremental energy assumptions (E).

**Lemma 3** (Passivity induces a skew/dissipative representation). *Let $H$ be strictly convex and twice differentiable in a neighborhood of $z$, with co-energy $e = \nabla H(z)$ and Hessian $G(z) = \nabla^2 H(z) \succ 0$. Assume $F$ is differentiable at $z$. Suppose the incremental energy conditions hold in a neighborhood of $z$, i.e., for all $z_1, z_2$ near $z$ with $e_i = \nabla H(z_i)$,*

$$(e_1 - e_2)^\top\big(F_{\mathrm{con}}(z_1) - F_{\mathrm{con}}(z_2)\big) = 0, \qquad (e_1 - e_2)^\top\big(F_{\mathrm{diss}}(z_1) - F_{\mathrm{diss}}(z_2)\big) \le 0. \tag{5}$$

*Then there exist matrices $J(z)$ and $R(z)$ with $J^\top = -J$ and $R^\top = R \succeq 0$ such that*

$$\frac{\partial F}{\partial z}(z) \;=\; \big(J(z) - R(z)\big)\, G(z). \tag{6}$$

*Proof.* Fix $z$ and a direction $\delta z$. Set $z_1 = z + \varepsilon\, \delta z$, $z_2 = z$ and define

$$\phi_{\mathrm{con}}(\varepsilon) := (e_1 - e_2)^\top\big(F_{\mathrm{con}}(z_1) - F_{\mathrm{con}}(z_2)\big), \quad \phi_{\mathrm{diss}}(\varepsilon) := (e_1 - e_2)^\top\big(F_{\mathrm{diss}}(z_1) - F_{\mathrm{diss}}(z_2)\big). \tag{7}$$

By the incremental energy conditions, $\phi_{\mathrm{con}}(\varepsilon) \equiv 0$ and $\phi_{\mathrm{diss}}(\varepsilon) \le 0$ for small $\varepsilon$. Differentiability of $H$ gives

$$e_1 - e_2 = \varepsilon\, G(z)\, \delta z + r_e(\varepsilon), \qquad \|r_e(\varepsilon)\| = o(\varepsilon), \tag{8}$$

and differentiability of $F_{\mathrm{con}}, F_{\mathrm{diss}}$ with the chain rule at $z$ yields

$$F_{\mathrm{con}}(z_1) - F_{\mathrm{con}}(z_2) = \varepsilon\, \frac{\partial F_{\mathrm{con}}}{\partial e}(z)\, G(z)\, \delta z + r_{\mathrm{con}}(\varepsilon), \tag{9}$$

$$F_{\mathrm{diss}}(z_1) - F_{\mathrm{diss}}(z_2) = \varepsilon\, \frac{\partial F_{\mathrm{diss}}}{\partial e}(z)\, G(z)\, \delta z + r_{\mathrm{diss}}(\varepsilon), \tag{10}$$

with $\|r_{\mathrm{con}}(\varepsilon)\|, \|r_{\mathrm{diss}}(\varepsilon)\| = o(\varepsilon)$. Writing $\delta e := G(z)\, \delta z$, we obtain the expansions

$$\phi_{\mathrm{con}}(\varepsilon) = \varepsilon^2\, \delta e^\top\Big(\tfrac{\partial F_{\mathrm{con}}}{\partial e}(z)\Big)\delta e + o(\varepsilon^2), \qquad \phi_{\mathrm{diss}}(\varepsilon) = \varepsilon^2\, \delta e^\top\Big(\tfrac{\partial F_{\mathrm{diss}}}{\partial e}(z)\Big)\delta e + o(\varepsilon^2). \tag{11}$$

Divide by $\varepsilon^2$ and let $\varepsilon \to 0$:

$$\delta e^\top\Big(\tfrac{\partial F_{\mathrm{con}}}{\partial e}(z)\Big)\delta e = 0, \qquad \delta e^\top\Big(\tfrac{\partial F_{\mathrm{diss}}}{\partial e}(z)\Big)\delta e \le 0 \quad (\forall\, \delta e). \tag{12}$$

Since $x^\top K x = x^\top \operatorname{Sym}(K)\, x$, the first equality forces $\operatorname{Sym}(\tfrac{\partial F_{\mathrm{con}}}{\partial e}(z)) = 0$; set $J(z) := \tfrac{\partial F_{\mathrm{con}}}{\partial e}(z)$ so $J^\top = -J$. The second inequality implies $\operatorname{Sym}(\tfrac{\partial F_{\mathrm{diss}}}{\partial e}(z)) \preceq 0$; define

$$R(z) := -\operatorname{Sym}\Big(\tfrac{\partial F_{\mathrm{diss}}}{\partial e}(z)\Big) \succeq 0, \qquad \widetilde{J}(z) := J(z) + \operatorname{Skew}\Big(\tfrac{\partial F_{\mathrm{diss}}}{\partial e}(z)\Big), \tag{13}$$

so that $\tfrac{\partial F}{\partial e}(z) = \widetilde{J}(z) - R(z)$ with $\widetilde{J}^\top = -\widetilde{J}$. Renaming $\widetilde{J}$ as $J$ and using $\tfrac{\partial e}{\partial z}(z) = G(z)$ yields

$$\frac{\partial F}{\partial z}(z) = \frac{\partial F}{\partial e}(z)\, \frac{\partial e}{\partial z}(z) = \big(J(z) - R(z)\big)\, G(z), \tag{14}$$

as claimed. $\qquad\qquad\square$

A.3   IDENTIFICATION OF $J$ WITH THE TOPOLOGY–INDUCED INTERCONNECTION

**Lemma 4** (Conservative interconnection is incidence–assembled). *Assume* (L), (P), *and* (O). *Then, up to permuting cochains within each degree and choosing orientation conventions, the conservative interconnection $J$ coincides with the topology–induced wiring assembled from the signed coboundaries $\{D_k\}$. Equivalently, there exists a degree–wise positive diagonal matrix $S$ (independent of state and time) such that, for all $k$,*

$$SJS^\top \text{ has off–diagonal } (k, k+1) \text{ blocks equal to } \begin{pmatrix} 0 & -D_k^\top \\ D_k & 0 \end{pmatrix}.$$

*In particular, $J$ is metric–free and determined solely by the mesh topology.*

*Proof.* By Lemma 1, any degree–$k$ output can involve only $\{\mathrm{Id}, A_k, D_{k-1}, -D_k^\top\}$. By Lemma 2, the same–degree terms ($\mathrm{Id}$ and $A_k$) are excluded by orientation oddness, so only $D_{k-1}$ and $-D_k^\top$ remain in the $k$–th row/column. Writing the $(k, k+1)$ and $(k+1, k)$ blocks as $-c_k D_k^\top$ and $d_k D_k$, skew–symmetry $J^\top = -J$ forces $c_k = d_k$. By (P), $c_k$ is uniform within each degree/type class; after fixing an orientation we may take $c_k > 0$. Collecting these positive degree–wise scalings into a diagonal $S$ yields the stated similarity form, which depends only on $\{D_k\}$ and not on the state or time. $\square$

Let $S = \mathrm{diag}(s_0 I, \ldots, s_d I)$ and set $J' := SJS^\top$. Then

$$J'_{k,k+1} = -(s_k s_{k+1} c_k)\, D_k^\top, \qquad J'_{k+1,k} = (s_k s_{k+1} c_k)\, D_k.$$

Choosing $s_k s_{k+1} = 1/c_k$ normalizes each adjacent pair to $\begin{pmatrix} 0 & -D_k^\top \\ D_k & 0 \end{pmatrix}$. Hence (up to cochain ordering, orientation gauge, and degree-wise rescaling) $J$ is the incidence-assembled interconnection built from $\{D_k\}$. $\square$

A.4   PROOF OF THEOREM 1

Fix a point $z$ where the Jacobian exists. By Lemma 3,

$$\frac{\partial F}{\partial z}(z) \;=\; \big(J(z) - R(z)\big) G(z), \qquad J(z)^\top = -J(z), \;\; R(z)^\top = R(z) \succeq 0, \qquad (15)$$

with $G(z) = \nabla^2 H(z) \succ 0$. By Lemma 4 (using (L), (P), and (O)), the conservative part $J(z)$ matches the incidence-assembled interconnection built from the signed coboundaries $\{D_k\}$, up to harmless conventions (reordering cochains, choosing orientations, and positive rescalings per degree). Concretely,

$$J(z) \;\sim\; \mathrm{blkdiag}\Big(\ldots, \begin{pmatrix} 0 & -D_k^\top \\ D_k & 0 \end{pmatrix}, \ldots\Big), \qquad (16)$$

where "$\sim$" means equality up to cochain ordering, orientation gauge, and degree-wise positive scalings.

Permutation equivariance (P) rules out interface-by-interface variability within a fixed degree/type pair, so any remaining scale is uniform per adjacent degree pair. In general these scales may depend on $z$, in which case $J$ need not be constant; the pointwise factorization above is the final statement.

*If, in addition, the degree-pair scales are state-independent* (geometry/material only), a fixed degree-wise rescaling of units absorbs them, yielding a constant, topology-only $J$ (up to ordering/orientation gauge). Under this additional assumption we obtain, locally,

$$\frac{\partial F}{\partial z}(z) \;=\; \big(J - R(z)\big) G(z), \qquad J^\top = -J, \;\; R(z)^\top = R(z) \succeq 0, \qquad (17)$$

with $J$ the incidence-assembled (topology-induced) interconnection built from $\{D_k\}$, proving Theorem 1 in the main text. State-/material-/geometry-dependency of the interconnection $J$ is discussed in the following remark.

**Remark: State-/Material-/Geometry-Dependent Interconnection.** Relaxing the topology–metric split to allow *state-dependence* (and materil-/geometry-dependent) in the conservative coupling does not change the pointwise structure. At any $z$ where the Jacobian exists, Lemmas 3 implies

$$\frac{\partial F}{\partial z}(z) = \big(J(z) - R(z)\big)G(z), \qquad J(z)^\top = -J(z), \ \ R(z)^\top = R(z) \succeq 0, \qquad (18)$$

and, under (L), (P), (O), each off–diagonal $(k, k{+}1)$ block of $J(z)$ must have the incidence form

$$J_{k,k+1}(z) = -c_k(z)\, D_k^\top, \qquad J_{k+1,k}(z) = c_k(z)\, D_k, \qquad (19)$$

with $c_k(z) > 0$. Here $c_k$ may depend on *state* as well as on *geometry/material*; it must be *orientation-even* (unchanged by degree-$k$ flips) to respect (O). To respect permutation equivariance (P), $c_k$ must be computed from type-shared, order-invariant local features (ML view: weight sharing over a degree/type pair), i.e.

$$c_k(\pi{\cdot}z) = c_k(z) \quad \text{for any degree/type–preserving permutation } \pi. \qquad (20)$$

If $c_k$ is *state-independent* (e.g., depends only on geometry/material), it can be absorbed by a fixed, degree-wise positive rescaling (a change of units): choose a block-diagonal $S = \text{diag}(\ldots, s_k I, s_{k+1} I, \ldots)$ with

$$s_k\, s_{k+1}\, c_k = 1 \qquad \text{for each adjacent pair } (k, k{+}1), \qquad (21)$$

so that

$$SJS^\top \text{ has } (k, k{+}1) \text{ blocks } \begin{pmatrix} 0 & -D_k^\top \\ D_k & 0 \end{pmatrix}. \qquad (22)$$

Under the associated change of variables $z = S\,\tilde{z}$ (equivalently, a degree-wise unit rescaling), the vector field and factors transform as

$$\tilde{F}(\tilde{z}) = S^{-1}F(S\tilde{z}), \qquad \tilde{J} = S^{-1}JS^{-T}, \qquad \tilde{R} = S^{-1}RS^{-T}, \qquad \tilde{G} = S^\top GS, \qquad (23)$$

and the Jacobian factorization is preserved:

$$\frac{\partial \tilde{F}}{\partial \tilde{z}}(\tilde{z}) = S^{-1}\Big(\frac{\partial F}{\partial z}(z)\Big)S = S^{-1}\big((J - R)G\big)S = (\tilde{J} - \tilde{R})\,\tilde{G}. \qquad (24)$$

Thus the normalization moves the degree-pair gains into $G/R$ while leaving the incidence wiring intact.

In contrast, if $c_k$ depends on $z$, the rescaling $S(z)$ would be state-dependent, and absorbing $c_k(z)$ would introduce extra Jacobian terms (by the product/chain rule), so the interconnection remains state-dependent:

$$\frac{\partial}{\partial z}\big(S(z)^{-1}F(z)\big) = S(z)^{-1}\frac{\partial F}{\partial z}(z) - S(z)^{-1}\Big(\frac{\partial S}{\partial z}(z)\, S(z)^{-1}\Big)F(z). \qquad (25)$$

Practically, one may either (i) restrict $c_k$ to be state-independent (geometry/material only), recovering a fixed wiring after absorption, or (ii) keep the incidence pattern fixed and *learn* coefficients $c_k(z)$ under the constraints above. In both cases the local reduction

$$\frac{\partial F}{\partial z}(z) = \big(J(z) - R(z)\big)G(z) \qquad (26)$$

holds pointwise; the difference is whether $c_k$ can be absorbed into a fixed energy metric. State-dependent example phenomena and their specific handling will be discussed in Appendix B.

**Generality and Sharpness.** The reduction is architecture-independent and applies to any dynamics $F$ whose Jacobian satisfies (L), (P), (O), and (E), yielding the pointwise factorization $\partial F/\partial z = (J(z) - R(z))G(z)$ with $J(z)$ incidence-assembled; under state-independent interconnection this further specializes to a constant, topology-only $J$. In addition, the hypotheses are essentially minimal: dropping (O) allows same-degree, orientation-insensitive terms (identity $I$ and unsigned adjacency $A_k$) to persist; dropping (P) permits interface-wise heterogeneity within a degree/type class (coefficients vary per interface), breaking equivariance and the clean topological assembly; dropping (L) admits nonlocal couplings that cannot be expressed on the incident matrix; relaxing (E) forfeits the $(J{-}R)G$ skew–dissipative split. Thus each assumption rules out a concrete failure mode, and together they deliver a strong structure theorem that separates what is fixed (topological wiring) from what is learnable (metric/dissipation), with a clear constant-$J$ specialization when appropriate. An empirical ablation study of each component is provided in Appendix D.7.

**Linear-Media Corollary.** If, in addition, $H(z) = \frac{1}{2}z^\top M^{-1}z$ with constant $M \succ 0$ and $F_{\text{diss}}(z) = -R\,e$ with state-independent $R^\top = R \succeq 0$, then $G \equiv M^{-1}$ and the dynamics reduce to

$$\dot{z} \;=\; (J - R)\, M^{-1} z, \qquad J^\top = -J, \;\; R^\top = R \succeq 0, \tag{27}$$

with $(M, R)$ carrying the metric and dissipation.

## B  EXTENSION OF MESHFT-NET FOR STATE-DEPENDENT INTERCONNECTION

**Motivation.** In continuum physics, energy-preserving couplings are often *state-dependent*: their strength is modulated by the physical states while remaining skew (doing no work). An important example is ideal MHD, in which the conservative interconnection depends on fields such as mass density and the magnetic field due to the underlying noncanonical Poisson-bracket structure (Morrison & Greene, 1980; Morrison, 1982). Although we focus on the state-independent conservative interconnection to clarify the basic structure of MeshFT and MeshFT-Net in this paper, the state-dependent case is essential for more realistic and general physical phenomena.

Nevertheless, with a straightforward extension—replacing the constant interconnection $J$ by its state-dependent counterpart $J(z)$ while retaining the same sparsity pattern and skew structure—MeshFT and MeshFT-Net readily accommodate state-dependent interconnections. We briefly outline this extension below.

**Pointwise Reduction.** At any state $z$ where the Jacobian exists, Lemmas 3 gives the local factorization

$$\frac{\partial F}{\partial z}(z) \;=\; \big(J(z) - R(z)\big)\, G(z), \qquad J(z)^\top = -J(z), \;\; R(z)^\top = R(z) \succeq 0, \tag{28}$$

with the *same sparsity pattern* for $J(z)$ (up to cochain ordering and orientation gauge). Concretely, each off–diagonal $(k, k{+}1)$ block has the incidence form

$$J_{k,k+1}(z) \;=\; -\,c_k(z)\,D_k^\top, \qquad J_{k+1,k}(z) \;=\; c_k(z)\,D_k, \quad c_k(z) > 0, \tag{29}$$

where $c_k(z)$ may depend on the current fields but must be unchanged under reversing the orientation convention of all degree-$k$ cells, and remain permutation-equivariant within each degree/type class. If $c_k$ is state-independent (geometry/material only), a fixed degree-wise rescaling absorbs $c_k$ into $G$, recovering a constant, topology-only $J$ (the main-text setting), as noted in the preceding remark.

**Time Stepping.** In practice, the dynamics given by Eq. 28 can be advanced with standard numerical schemes. Note that there is a trade-off between structure preservation and computational cost, and the optimal choice is problem-dependent; a full comparison lies beyond our scope, so we list minimal options and trade-offs. (i) *Midpoint discrete gradient (energy balanced):* evaluate $(J, R)$ at the midstate $\bar{z} = \frac{1}{2}(z^{n+1} + z^n)$ and use a discrete gradient $\bar{\nabla}H$ with $H(z^{n+1}) - H(z^n) = \bar{\nabla}H^\top(z^{n+1} - z^n)$; this preserves the energy inequality exactly (Gonzalez, 1996). (ii) *Poisson–dissipative splitting (Strang):* freeze $J, R$ at $z^\star$ (e.g., $\bar{z}$) and compose a symplectic Hamiltonian substep with an implicit/discrete-gradient dissipative substep; this is second order and structure preserving (Strang, 1968). (iii) *Fully explicit baseline:* an RK2/Heun step with $J, R$ frozen per stage is simple to implement, but does not enforce the exact energy balance.

*Computational note:* Methods (i)–(ii) are implicit and therefore more expensive per step, trading cost for stability and exact structure preservation, whereas (iii) is cheaper but forfeits exact energy balance.

**Topology and State-Dependent Coefficients.** In the case of state-dependent interconnection, the wiring is assembled from signed incidences $\{D_k\}$ (topology only), while state-dependence enters only via the scalar gains $c_k(z)$ per degree pair. Consequently, identification/learning reduces to estimating these (possibly state-/geometry-/material-dependent) gains $c_k(z)$; the incidence wiring $\{D_k\}$ is fixed by topology and need not be learned. This keeps the main reduction intact: $J(z)$ retains the same incidence sparsity and blockwise skew structure, so the theorem's significance is unaffected.

## C  ELECTROMAGNETISM EXAMPLE

As an intuitive example, we revisit electromagnetism though the lens of Theorem 1.

### C.1  RECOVERING SOURCE-FREE MAXWELL FROM TOPOLOGY-FIXED INCIDENCE STRUCTURE

On an oriented mesh with edge–face incidence $D_1$, the constitutive maps are the (degree–wise) SPD Hodge stars

$$H = \star_{\mu^{-1}} B, \qquad E = \star_{\varepsilon^{-1}} D, \tag{30}$$

where $B$ is the magnetic flux density and $H$ the magnetic field intensity; $D$ is the electric flux density and $E$ the electric field. The operator $\star_\kappa$ denotes the (discrete) Hodge star composed with the material tensor $\kappa$ (e.g., $\mu^{-1}$, $\varepsilon^{-1}$).

So, stacking $z = (B, D)$ and $e = (H, E)$, one has

$$e = \underbrace{\begin{pmatrix} \star_{\mu^{-1}} & 0 \\ 0 & \star_{\varepsilon^{-1}} \end{pmatrix}}_{G_\theta^{-1}} z. \tag{31}$$

The conservative Maxwell update (no sources) is

$$\dot{B} = -D_1 E \quad \text{(Faraday)}, \qquad \dot{D} = D_1^\top H \quad \text{(Ampère)}, \tag{32}$$

which compactly reads

$$\dot{z} = J e = \underbrace{\begin{pmatrix} 0 & -D_1^\top \\ D_1 & 0 \end{pmatrix}}_{J} \underbrace{G_\theta^{-1}}_{\text{SPD}} z. \tag{33}$$

i.e., the special case of our update $\dot{z} = (J - R_\theta)G_\theta^{-1} z$ with $R_\theta \equiv 0$. Here $J$ is fixed entirely by signed incidences (mesh topology and orientation), while material/geometry enter only through the learned SPD metric blocks $G_\theta^{-1}$ (the constitutive law $\varepsilon^{-1}, \mu^{-1}$).

**Maxwell Structure as a Consequence.**  Under the requirements used in the main text, the MeshFT thus recovers the source–free Maxwell update on arbitrary oriented meshes without explicitly postulating the Maxwell equations: orientation–aware incidences determine the conservative wiring $J$, and the *only* trainable physical freedom is the degree–wise SPD Hodge blocks in $G_\theta^{-1}$ (the material law). Unlike residual–based designs, we do not enforce a PDE loss; the structure rules out non-physical couplings by construction and focuses learning on the constitutive mapping.

**Note on Learning Freedom.**  The structural constraint above *does not* fix any particular formula for the discrete Hodge stars; it only requires the degree-wise blocks of $G_\theta^{-1}$ to be SPD (and, if desired, local and permutation-equivariant). Thus the incidence wiring $J$ is fixed by topology, whereas all geometry/material dependence resides in $G_\theta^{-1}$—precisely the learnable, unconstrained degrees of freedom (e.g., metric structure and spatially varying $\varepsilon^{-1}$, $\mu^{-1}$) that can be identified from data without violating the topological structure.

### C.2  DEGENERACY FROM TOPOLOGY AND ELIMINATION OF SPURIOUS MODES

Because DEC encodes the chain–complex identity $D_{k+1}D_k = 0$, the interconnection $J$ above is *rank–deficient* (degenerate): it has a nontrivial kernel aligned with topological constraints. Two immediate invariants on closed, source–free domains are

$$D_2 \dot{B} = -D_2 D_1 E = 0, \tag{34}$$

$$D_0^\top \dot{D} = D_0^\top D_1^\top H = (D_1 D_0)^\top H = 0, \tag{35}$$

expressing discrete $\nabla \cdot B = 0$ and the charge–free $\nabla \cdot D = 0$, respectively. These invariants do not depend on the metric maps $G_\theta^{-1}$ and hold to machine precision, thereby suppressing non-physical (spurious) modes such as fake magnetic monopoles or artificial charge accumulation. Crucially, these invariants are not a post hoc regularizer but a direct consequence of the physical constraints structurally enforced by MeshFT. This topological guarantee is a major difference from methods that learn physical dynamics on meshes, which typically cannot preserve such invariants exactly.

C.3  RELATION TO SYMPLECTIC–ODE (NONDEGENERATE) FORMULATIONS

Symplectic ODE integrators preserve a *global* symplectic two-form on a finite-dimensional (gauge-reduced) phase space—e.g., in canonical coordinates $(q, p)$ with $\Omega = \sum_i dp_i \wedge dq_i$, and dynamics $X_H = \Omega^{-1}dH$. This is structure-preserving, but $\Omega$ by itself does *not* encode the mesh incidence operators or the exactness $D_{k+1}D_k = 0$. Consequently, discrete Gauss-type constraints (e.g., divergence constraints) are not automatically invariant: they must be enforced by projection/cleaning or by special constraint-preserving discretizations; otherwise spurious modes can emerge over long horizons even though the global symplectic form is conserved.

By contrast, fixing the interconnection via DEC (the $J$ matrix) *builds the topology into the dynamics*: $J$ is assembled from incidences, so the relations like $D_{k+1}D_k = 0$ are hard-wired, and the corresponding constraint subspaces are preserved mechanically. This cleanly separates *topology/interconnection* ($J$) from *metric/constitutive content* (Hodge stars), which can then carry geometry/material information.

# D  DETAILS OF EXPERIMENTS AND ADDITIONAL RESULTS

**Link to Theorem 1 (notation).**  In this section, we use the standard FEM symbols $M$ (0-form Hodge on nodes) and $W$ (1-form Hodge on edges) as the concrete *metric blocks* of Theorem 1. Throughout the experiments given in this paper, we assumed $M$ and $W$ are *state-independent*. With canonical packaging $z = (q, p) \in C^k \oplus C^k$, we define the canonical momentum as $p := M\dot{q}$ (so that the kinetic energy is $\frac{1}{2}\dot{q}^\top M\dot{q} = \frac{1}{2}p^\top M^{-1}p$). Then

$$H(z) = \tfrac{1}{2}q^\top K q + \tfrac{1}{2}p^\top M^{-1}p, \qquad K = D_0^\top W D_0, \tag{36}$$

so

$$e = \nabla H(z) = \begin{pmatrix} Kq \\ M^{-1}p \end{pmatrix} = G^{-1}z, \qquad G^{-1} = \begin{pmatrix} K & 0 \\ 0 & M^{-1} \end{pmatrix}. \tag{37}$$

The conservative interconnection is the canonical symplectic matrix $J = \left(\begin{smallmatrix} 0 & I \\ -I & 0 \end{smallmatrix}\right)$, and the dynamics is $\dot{z} = (J - R(z))e$ with $R(z)^\top = R(z) \succeq 0$. (Equivalently, under the mixed packaging $z = (x^{(k)}, x^{(k+1)}) \in C^k \oplus C^{k+1}$, $J = \left(\begin{smallmatrix} 0 & -D_0^\top \\ D_0 & 0 \end{smallmatrix}\right)$; absorbing $D_0$ into $K = D_0^\top W D_0$ yields the canonical form above.)

## D.1  ANALYTIC PLANE-WAVE ON PERIODIC 2D MESHES

**Meshes and Geometry.**  On a torus, we use (i) a periodic axis-aligned grid and (ii) a Delaunay triangulation. We report results on regular grid or Delaunay mesh at a nominal $32 \times 32$ resolution. Node dual areas $V_0$ come from cell/barycentric areas; edge weights $V_1^{-1}$ use nonnegative cotangent weights computed with periodic minimum–image distances, with small quantile floors to avoid degenerate areas/edges. Edge features are $(\Delta x, \Delta y, \|e\|)$; node features are $(x, y, V_0)$.

**Data Generation.**  Each training pair $(z_t, z_{t+\Delta t})$ uses the canonical state $z = (q, p)$ with a single traveling wave

$$q(t) = a\sin(k^\top x - \omega t + \phi), \qquad \omega = c\|k\|, \qquad p = V_0\dot{q}(t),$$

where $x$ are nodal coordinates on the periodic box $[0, L]^2$ and $V_0$ is the node volumes. For each sample we draw a wavenumber $k = \frac{2\pi}{L}(k_x, k_y)$ with $k_x = s_x U_x, U_x \sim \text{Unif}\{1, \cdots, 4\}, s_x \in \{\pm 1\}$ equiprobable, and $k_y = s_y U_y, U_y \sim \text{Unif}\{0, \cdots, 4\}, s_y \in \{\pm 1\}$; the zero mode $(k_x, k_y) = (0, 0)$ is excluded. Independently, we sample the phase $\phi \sim \text{Unif}[0, 2\pi)$, amplitude $a \sim \text{Unif}[0.5, 1.5]$, and start time $t_0 \sim \text{Unif}[0, 2\pi)$. We then set

$$z_t = (q(t_0), V_0\dot{q}(t_0)), \qquad z_{t+\Delta t} = (q(t_0 + \Delta t), V_0\dot{q}(t_0 + \Delta t)).$$

Randomness is controlled via synchronized seeds; splits and batches are identical across models.

**Hodge Parametrization.**  MeshFT-Net fixes the interconnection $J$ via the signed incidences $\{D_k\}$ and *learns only the metric*. A small geometry–conditioned Hodge maps node/edge features to positive stars:

$$M_i = V_{0,i}\,\sigma(\phi_{\text{node}}(x_i, y_i, V_{0,i})), \qquad W_e = V_{1,e}^{-1}\,\sigma(\phi_{\text{edge}}(\Delta x_e, \Delta y_e, |e|)),$$

Table 6: One-step MSE and energy drift by models trained on sufficient amount of training data for the regular grid data. Lower is better; drift closer to $0$ is better. All numbers are computed on held-out validation data. Mean $\pm$ s.d. over 3 seeds are reported.

| Model | One-step MSE | Energy drift $\Delta E/E_0$ |
|---|---|---|
| MeshFT-Net | $\mathbf{1.4 \times 10^{-7} \pm 6.1 \times 10^{-9}}$ | $\mathbf{7.2 \times 10^{-3} \pm 1.4 \times 10^{-3}}$ |
| MGN | $1.2 \times 10^{-6} \pm 2.5 \times 10^{-7}$ | $12.0 \pm 12.6$ |
| MGN-HP | $5.7 \times 10^{-4} \pm 1.8 \times 10^{-5}$ | $7.3 \pm 3.1$ |
| HNN | $4.3 \times 10^{-7} \pm 4.7 \times 10^{-7}$ | $0.63 \pm 0.25$ |

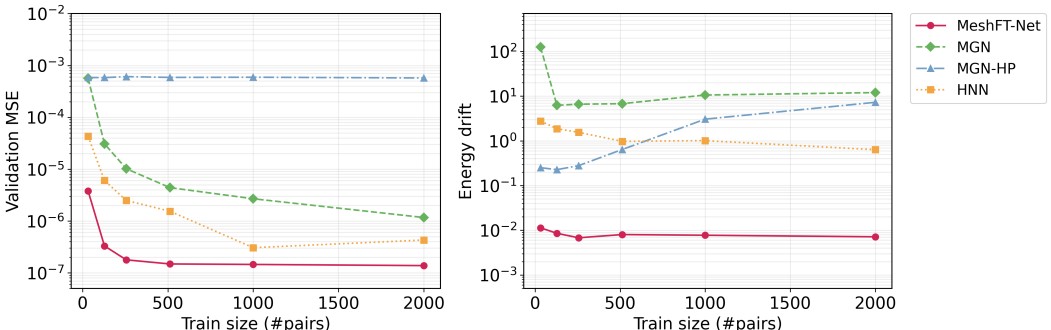

Figure 4: Relationship between the size of training dataset and one-step MSE (left) and rollout energy drift (right) for random delaunay mesh.

with $\sigma = \mathrm{softplus}$. Here $\phi_{\mathrm{node}}$ and $\phi_{\mathrm{edge}}$ are *state–independent* MLPs (2 layers, width 64) that take standardized geometry as input and output a scalar log–scale; applying $\sigma$ ensures $M_i > 0$ and $W_e > 0$. This geometry-conditioned Hodge is used throughout the experiments below, unless otherwise stated.

**Metrics and Hyperparameters.** For evaluation, a shared theory Hodge $(M, W) = (V_0, c^2 V_1^{-1})$ defines the physical norm used for relative error and for energy-drift over open-loop rollouts ($\Delta t = 0.002$, $T = 200$). Training runs for 10 epochs with a mini-batch size of 8 on 2000 training pairs and 256 validation pairs. The wave speed in the analytic solution is set to $1.0$. MGN and MGN-HP use hidden width 64 with 4 message-passing layers; the network for predicting Hamiltonian in MGN-HP is configured identically (hidden 64, 4 layers). The HNN likewise uses hidden width 64 with 4 layers. The other exact hyperparameters are provided in the released code.

**Additional Results.** For completeness, in addition to the regular-grid results reported in the main text (Sec. 5.1), this appendix presents the same analytic plane-wave benchmark on periodic Delaunay triangulations under identical training protocols. The Delaunay results corroborate our main findings—MeshFT-Net maintains the lowest error and near-zero drift, while the baselines exhibit substantially larger drift—with detailed numbers and visualizations provided in Table 6 and Fig. 4.

### D.2 Dissipative Benchmark

**Data Generation.** We benchmark *damped* plane-waves on a periodic 2D grid, using the canonical packing $x = (q, p)$. Samples are generated as

$$q(x, t) = A\, e^{-\gamma t} \sin(k^\top x - \omega t + \phi), \qquad \gamma \sim \mathrm{Unif}\,[0.01, 0.1],$$

and we form one-step pairs $(x_t, x_{t+\Delta t})$ on a periodic $32 \times 32$ grid over $[0, 1]^2$, with $\Delta t = 0.002$, wave speed $c = 1.0$, and integer wavenumbers up to 6 (excluding the zero mode). The other parameters are same as the analytic plane-wave benchmark shown in Appendix D.1.

**Models and Integration.** Time stepping mirrors the conservative structure: MeshFT-Net uses KDK (with *exact half–step damping*, i.e., Strang split), HNN uses the same KDK-with-damping scheme,

and MGN/MGN–HP are wrapped in a symmetric KDK integrator applied to their learned vector field $v(x)$. For MeshFT-Net and HNN we use nodewise Rayleigh dissipation: $\dot{p} = -Kq - Rp$ with rates $r_i \geq 0$ (equivalently $R = \mathrm{diag}(r) \succeq 0$). Concretely, we use *GammaInferNet*—a small per-node damping MLP (message passing; width 64, 2 layers)—that ingests node/edge features, outputs raw rates $\tilde{r} \in \mathbb{R}$, and sets $r = \mathrm{softplus}(\tilde{r}) \geq 0$. In continuous time the momentum channel obeys $\dot{p} = -Kq - Rp$, and in our Strang-split KDK step we apply the exact exponential decay over a step $\Delta t$: $p \leftarrow e^{-\frac{\Delta t}{2} R} p$ (before and after the conservative pass). MGN/MGN–HP *do not* receive an explicit damping operator and must infer dissipation from data.

**Metrics and Hyperparameters.** We report (i) one–step MSE; and (ii) normalized energy drift over open-loop rollouts ($T = 200$) comparing with the true trajectory including the dissipation. Training runs for 20 epochs with a mini-batch size of 16 on 4000 training pairs and 256 validation pairs. MGN and MGN-HP use hidden width 64 with 4 message-passing layers; the network for predicting Hamiltonian in MGN-HP is configured identically (hidden 64, 4 layers). The HNN likewise uses hidden width 64 with 4 layers. The other exact hyperparameters are provided in the released code.

### D.3 Physically-Consistency Benchmark

**Meshes and Geometry.** We use periodic axis–aligned grids on the torus $[0, 1)^2$. The grid is $32{\times}32$, which is same as the analytic plane-wave benchmark shown in Appendix D.1.

**Data Generation.** Training pairs $(z_t, z_{t+\Delta t})$ come from analytic plane-waves $q(t) = a\sin(k^\top x - \omega t + \phi)$, with $\omega = c\|k\|$ and $c = 1.0$. Per sample we draw integer wavenumbers up to 6 (exclude the zero mode), a phase $\phi \sim \mathrm{Unif}[0, 2\pi]$, amplitude $a \sim \mathrm{Unif}[0.5, 1.5]$, and start time $t_0 \sim \mathrm{Unif}[0, 2\pi]$.

**Metrics and Hyperparameters.** All physics diagnostics are computed with a shared *theory* Hodge, $(M, W) = (V_0, c^2 V_1^{-1})$, to ensure fair comparison, independent of a model's internal parameterization. The details of metrics for physical consistency are summarized in Table 7. Each hyperparameters are set as $\Delta t = 0.002$, $T = 200$; epochs $= 10$, batch $= 16$, train size $= 4000$, val size $= 256$. The other exact hyperparameters are provided in the released code.

In addition to physics diagnostics, we evaluate model–PDE agreement at the *vector-field* level and short-horizon behavior, using the same theory Hodge. (i) *Vector-field alignment*: at a batch of states $z$ we compute the PDE field $v_{\mathrm{PDE}}(z) = [M^{-1}p, -Kq]$ and the model field $v_{\mathrm{model}}(z)$, and report the cosine similarity and the relative $L_2$ error $\|v_{\mathrm{model}} - v_{\mathrm{PDE}}\|/\|v_{\mathrm{PDE}}\|$. (ii) *Short-horizon rollout*: from the same initial states we roll out $T_{\mathrm{short}}{=}16$ steps and report the final relative error. (iii) *Amplitude/phase recovery*: for the predicted field at $t = t_0 + T_{\mathrm{short}}\Delta t$, fit $q(x) \approx a\sin(kx) + b\cos(kx)$ ($2{\times}2$ least squares) and report amplitude error $|\hat{A} - A|/|A|$ with $\hat{A} = \sqrt{a^2 + b^2}$, and phase error $|\hat{\phi} - \phi_{\mathrm{eff}}|$ in degrees with $\phi_{\mathrm{eff}} = \phi - \omega t$.

### D.4 OOD Generalization

**Data generation.** We use periodic 2D wave dynamics. Training and test pairs $(z_t, z_{t+\Delta t})$ are sampled from analytic plane-waves $q(x, t) = a\sin(k^\top x - \omega t + \phi)$ with $\omega = c\|k\|$. For each sample we draw integer wavenumbers up to 3 (excluding the zero mode). The other parameters are same as the analytic plane-wave benchmark shown in Appendix D.1. Training uses one–step teacher forcing. We generate three OOD settings by changing the test distribution relative to training: (i) **frequency** ($k_{\mathrm{test}} > k_{\mathrm{train}}$), (ii) **resolution** (coarse→fine grid), (iii) **parameter** ($c_{\mathrm{test}} \neq c_{\mathrm{train}}$).

**Metrics and Hyperparameters.** All long-horizon errors and energy drift are evaluated with a theory Hamiltonian: on grids we use $(M, W) = (V_0, c^2 I_e)$. We report (i) one-step MSE on the test distribution; and (ii) normalized energy drift over the trajectory.

Training is performed on a $32{\times}32$ periodic grid with time step $\Delta t_{\mathrm{train}} = 0.004$, and wave speed $c_{\mathrm{train}} = 1.0$, using 4000 training pairs, batch size 16, and 10 epochs. Testing uses $\Delta t_{\mathrm{test}} = 0.004$, $T = 200$, and 512 test pairs; scenario-specific shifts are: (a) *frequency*—test_kmax = 6; (b) *resolution*—a finer $64{\times}64$ grid; (c) *parameter*—wave speed $c_{\mathrm{test}} = 1.4$. MGN and MGN-HP use hidden width 64 with 4 message-passing layers; the network for predicting Hamiltonian in MGN-HP is configured identically (hidden 64, 4 layers). The HNN likewise uses hidden width 64 with 4 layers. The other exact hyperparameters are provided in the released code.

Table 7: Details of physics metrics (computed on closed-loop rollouts with the shared *theory* Hodge $(M, W) = (V_0, c_{\text{speed}}^2 V_1^{-1})$). Frames are indexed $t = 0, \ldots, T$ (initial included).

| Diagnostic | Physical meaning and exact computation |
|---|---|
| Dispersion | Wave speed from the phase of $q$ at wavenumber $k$; estimate $\hat{\omega}$ from weighted phase increments and set $\hat{c} = \hat{\omega}/\|k\|$. Reported as relative absolute errors vs. ground truth. |
| Canonical consistency | Check $p \approx M\dot{q}$ using central differences (sum over interior times $t = 1, \ldots, T{-}1$): $\dot{q}_t \approx \dfrac{q_{t+1} - q_{t-1}}{2\Delta t}$, $p_t^{\text{mid}} = \frac{1}{2}(p_{t+1} + p_{t-1})$. Reported $$\frac{\sum_{t=1}^{T-1} \| p_t^{\text{mid}} - M\dot{q}_t \|_2^2}{\sum_{t=1}^{T-1} \| M\dot{q}_t \|_2^2}.$$ |
| PDE residual | Discrete wave equation mismatch with $K = D_0^\top W D_0$ (again $t = 1, \ldots, T{-}1$): $\ddot{q}_t \approx \dfrac{q_{t+1} - 2q_t + q_{t-1}}{(\Delta t)^2}$, $r_t = M\ddot{q}_t + Kq_t$. Reported $$\frac{\sum_{t=1}^{T-1} \|r_t\|_2^2}{\sum_{t=1}^{T-1} \left( \|M\ddot{q}_t\|_2^2 + \|Kq_t\|_2^2 \right)}.$$ |
| Equipartition | Time-averaged kinetic/potential balance: $T_t = \frac{1}{2} p_t^\top M^{-1} p_t$, $U_t = \frac{1}{2} q_t^\top K q_t$ ($= \frac{1}{2} (Bq_t)^\top W (Bq_t)$), $\langle T \rangle = \frac{1}{T+1} \sum_{t=0}^{T} T_t$, $\langle U \rangle = \frac{1}{T+1} \sum_{t=0}^{T} U_t$, and reported $$\frac{|\langle T \rangle - \langle U \rangle|}{\langle T \rangle + \langle U \rangle}.$$ |
| Momentum | Normalized range of total momentum over time: $m_t = \sum_{n=1}^{N} p_t[n]$. Reported $$\frac{\max_{0 \leq t \leq T} m_t - \min_{0 \leq t \leq T} m_t}{\frac{1}{T+1} \sum_{t=0}^{T} \sum_{n=1}^{N} |p_t[n]|}.$$ |

**Additional Results.** The full results are provided in Table 8.

D.5 VALIDATION ON REAL PHYSICS FIELD DATA FROM *THE WELL*

**Data generation.** We evaluate the *acoustic_scattering_discontinuous* subset of *The Well* (2D acoustics; near-Hamiltonian with discontinuous media). Fields are placed on a Cartesian grid of $256 \times 256$ nodes over $[-1, 1] \times [-1, 1]$ with mixed boundary conditions—*reflecting* in $x$ and *open* in $y$. From each short sequence we build one–step pairs $(z_t, z_{t+\Delta t})$ in canonical packing $z = (q, p)$, where $q$ is pressure and $p = M\dot{q}$ with node mass $M = V_0$ (cell area). The $p$ used for training is discretely calculated based on $q$. The dataset time step is $\Delta t = 2/101 \approx 0.0198$. To emulate open $y$-boundaries consistently across time stepping, we apply a light *sponge* after each update: $p \leftarrow p - \gamma_{\text{bias}}(y)\, p\, \Delta t$, with the bias ramped over the top/bottom $8\%$ of the domain.

**Hyperparameters.** Training follows the runner settings: 5 epochs, batch size 2, and 800 steps/epoch (validation every 50 steps). For stability, MeshFT-Net employs CFL-based *substepping* with target Courant number 0.5; on $256 \times 256$ and $\Delta t \approx 0.0198$ this yields about 8 substeps per data step. The other exact hyperparameters are provided in the released code.

**Parameterization** We parameterize the discrete Hodge operators with compact, data-driven MLPs that are evaluated *locally* on nodes and edges.

Table 8: OOD generalization across three scenarios. Lower is better. Bold indicates the best and underline the second best in each column. Mean $\pm$ s.d. over 3 seeds are reported.

(a) Frequency extrapolation ($k_{\text{test}} > k_{\text{train}}$)

| Model | One-step MSE | TSMSE | Drift |
|---|---|---|---|
| MeshFT-Net | $\mathbf{4.1 \times 10^{-6} \pm 2.6 \times 10^{-6}}$ | $\mathbf{0.18 \pm 0.11}$ | $\mathbf{5.1 \times 10^{-3} \pm 1.2 \times 10^{-3}}$ |
| MGN | $\underline{2.3 \times 10^{-5} \pm 1.8 \times 10^{-5}}$ | $9.0 \pm 11.9$ | $>100$ |
| MGN-HP | $6.5 \times 10^{-3} \pm 4.0 \times 10^{-3}$ | $\underline{0.62 \pm 0.30}$ | $\underline{0.16 \pm 9.3 \times 10^{-2}}$ |
| PI-MGN | $1.0 \times 10^{-3} \pm 1.4 \times 10^{-3}$ | $>100$ | $>100$ |
| HNN | $4.8 \times 10^{-5} \pm 5.9 \times 10^{-5}$ | $1.5 \pm 1.4$ | $0.45 \pm 0.53$ |
| FNO | $3.5 \times 10^{-4} \pm 7.0 \times 10^{-4}$ | $1.9 \pm 1.6$ | $2.7 \pm 2.2$ |
| GraphCON | $8.7 \times 10^{-5} \pm 7.7 \times 10^{-5}$ | $69.0 \pm 1.1 \times 10^2$ | $96.5 \pm 1.2 \times 10^2$ |

(b) Parameter shift ($c_{\text{test}} \neq c_{\text{train}}$)

| Model | One-step MSE | TSMSE | Drift |
|---|---|---|---|
| MeshFT-Net | $5.8 \times 10^{-6} \pm 5.9 \times 10^{-6}$ | $\underline{0.71 \pm 0.39}$ | $\mathbf{0.17 \pm 8.0 \times 10^{-3}}$ |
| MGN | $\mathbf{3.1 \times 10^{-6} \pm 2.6 \times 10^{-6}}$ | $1.7 \pm 1.1$ | $24.5 \pm 25.2$ |
| MGN-HP | $3.2 \times 10^{-3} \pm 2.7 \times 10^{-3}$ | $\mathbf{0.59 \pm 0.29}$ | $0.25 \pm 0.20$ |
| PI-MGN | $2.6 \times 10^{-4} \pm 2.9 \times 10^{-4}$ | $>100$ | $>100$ |
| HNN | $8.7 \times 10^{-6} \pm 1.1 \times 10^{-5}$ | $0.80 \pm 0.44$ | $\underline{0.24 \pm 0.12}$ |
| FNO | $\underline{5.7 \times 10^{-6} \pm 4.5 \times 10^{-6}}$ | $0.96 \pm 0.56$ | $1.6 \pm 2.0$ |
| GraphCON | $2.6 \times 10^{-4} \pm 2.9 \times 10^{-4}$ | $>100$ | $>100$ |

(c) Resolution transfer ($32{\times}32 \to 64{\times}64$)

| Model | One-step MSE | TSMSE | Drift |
|---|---|---|---|
| MeshFT-Net | $\mathbf{7.0 \times 10^{-7} \pm 5.4 \times 10^{-7}}$ | $\mathbf{5.1 \times 10^{-2} \pm 2.5 \times 10^{-2}}$ | $\mathbf{2.9 \times 10^{-3} \pm 9.2 \times 10^{-4}}$ |
| MGN | $8.6 \times 10^{-4} \pm 7.0 \times 10^{-4}$ | $0.53 \pm 0.34$ | $3.4 \pm 2.7$ |
| MGN-HP | $1.5 \times 10^{-3} \pm 1.3 \times 10^{-3}$ | $0.58 \pm 0.29$ | $5.6 \pm 4.4$ |
| PI-MGN | $8.6 \times 10^{-4} \pm 6.7 \times 10^{-4}$ | $>100$ | $>100$ |
| HNN | $8.6 \times 10^{-4} \pm 7.0 \times 10^{-4}$ | $0.36 \pm 0.30$ | $2.0 \pm 0.47$ |
| FNO | $\underline{3.2 \times 10^{-5} \pm 3.7 \times 10^{-5}}$ | $>100$ | $>100$ |
| GraphCON | $8.9 \times 10^{-4} \pm 6.7 \times 10^{-4}$ | $\underline{0.32 \pm 0.20}$ | $\underline{0.71 \pm 0.13}$ |

For each node $i$, we predict a positive mass scale $\rho_i > 0$ with a small node-MLP fed by geometric/context features, including coordinates $(x_i, y_i)$ and cell area $V_{0,i}$. We then set

$$M_i = \rho_i V_{0,i}, \qquad \log \rho_i = \tanh\big(\phi_{\text{node}}(\cdot)\big),$$

where $\phi_{\text{node}}$ is a two-layer MLP (width 32).

For each edge $e = (i, j)$, we assemble geometric edge features $(\Delta x_e, \Delta y_e, |e|)$ and feed them to an edge-MLP. The edge-MLP outputs a positive scale $\sigma_e > 0$, and we define

$$W_e = \sigma_e V_{1,e}^{-1}, \qquad \log \sigma_e = \tanh\big(\phi_{\text{edge}}(\cdot)\big),$$

with $\phi_{\text{edge}}$ a two-layer MLP (width 32). Here $V_{1,e}$ denotes the (diagonal) discrete Hodge star on edges (primal 1-forms). On a regular grid we set $V_{1,e} = 1$ (hence $V_{1,e}^{-1} = 1$); on irregular meshes $V_1$ is precomputed from geometry. All other hyperparameters follow the released code.

**Visualization.** From a validation sequence we form the canonical initial state $z_0 = [q_0, p_0]$. We then perform an open-loop $K$-step rollout with the symplectic KDK step, $\hat{z}^{j+1} = \text{MeshFT-Net}_{\Delta t}(\hat{z}^j)$ for $j = 0, \ldots, K-1$, always feeding the *predicted* state into the next step. We set $K = 48$ and, in the main text, display five representative snapshots at frames 1, 12, 24, 36, and 48 (start/midpoints/end). At each step we record the predicted pressure $\hat{q}_j$ and render GT vs. MeshFT-Net contours side-by-side (using a color scale fixed by the GT frames).

### D.6 DISCUSSION ON NONLINEAR ADVECTION-DIFFUSION

**Data generation.** We use a periodic 2D advection–diffusion setting on $[0,1]^2$. Training and validation pairs $(q_t, q_{t+\Delta t})$ are produced by one explicit Euler step of

$$\frac{\partial q}{\partial t} + \nabla\cdot\big((a + \beta q)\, q\big) = \kappa\,\Delta q$$

with fixed parameters $a = (7, -4)$, $\kappa = 0.05$, $\beta = 0.8$, and $\Delta t = 5 \times 10^{-5}$ or $1 \times 10^{-3}$. Initial conditions are Gaussian blobs with random center/width and amplitude. We generate 4000 or 400 training and 256 validation pairs on a $64 \times 64$ grid.

**Parametrizations and integrators.** We use a MeshFT-Net on a periodic grid (nodes $N$, edges $E$). The state is $z = (q, r)$, where $q \in \mathbb{R}^N$ and $r \in \mathbb{R}^E$. Two small MLPs (width 64, 2 layers, SiLU, spectral norm) predict diagonal factors $G$ and $R$. Advection is advanced by an energy–preserving Cayley half–step; diffusion uses Crank–Nicolson with fixed–iteration conjugate gradients (12 iterations, tolerance $10^{-8}$).

**Metrics and hyperparameters.** We report long–horizon metrics over $T{=}500$ TSMSE and mass drift. The mass drift uses the discrete mass $m_t = \sum_i V_{0,i} q_i(t)$ and is reported as the net relative change $|m_T - m_0|/|m_0|$. Also, no explicit mass–conservation penalty is applied.

### D.7 ABLATIONS ON TOPOLOGY, ORIENTATION, AND METRIC STRUCTURE

**Objective.** Guided by Theorem 1, we test which structural assumptions matter in practice. Recall that locally $\frac{\partial F}{\partial z}(z) = (J - R(z))\, G(z)$ with $J^\top = -J$ assembled from the signed incidences $\{D_k\}$ (topology), $R(z)^\top = R(z) \succeq 0$ (dissipation), and $G(z) \succ 0$ (metric/constitutive). Our ablations selectively violate these ingredients while keeping data, optimizer, supervision, and the integrator fixed.

**Common setup.** All models are trained with one-step teacher forcing on analytic plane-wave pairs $x = [q, p]$ ($p = M\,\dot{q}$) on a $32 \times 32$ torus with step $\Delta t = 0.002$. Long-horizon evaluation uses $T = 200$ steps. For fair comparison, all rollout metrics use a *common* theory Hodge.

**Metrics.** (1) one-step MSE; (2) normalized energy drift $\frac{|E_t - E_0|}{|E_0|}$ over the rollout; (3) energy injection $E_{\text{inj}} = \sum_{t=0}^{T-1} \max(E_{t+1} - E_t, 0)\big/(|E_0| + \varepsilon)$; (4) momentum variation: time variation of $\sum_n p_n$ on the torus, normalized by the mean $\ell_1$ amplitude, as in the physics-consistency test in Table 3. Lower is better for all.

**Compared Variants.** We consider the following variants for ablation study. The other mplementation details are provided in the released code. **MeshFT-Net (structured baseline).** $J$ is assembled from the signed incidences $\{D_k\}$ (topology + orientation), the metric is positive ($G \succ 0$), and $R \equiv 0$ (conservative). *No assumptions are violated*; energy and momentum are conserved up to integrator error, and dispersion is correct. **No-Orientation (orientation dropped).** Replace the signed incidence by an orientation–even map, violating (O). The resulting interconnection is no longer skew, $J^\top \neq -J$, so the conservative pass can inject/remove energy; *expect* systematic energy drift and spurious momentum leakage. **Scrambled-Topology (topology broken).** Randomly re–pair the node–edge incidences so that $J$ is *not* the topology–assembled one in Theorem 1 and interface locality (L) is broken. Algebraic skew of $J$ is retained by construction, but under the common *theory* energy for evaluation, *expect* incorrect modal coupling and large long–horizon drift. **Indefinite-Metric (metric positivity broken).** Keep the signed topology (orientation preserved) but allow the edge–space weights to be signed, violating $G \succeq 0$. *Expect* nonphysical energy growth, positive energy injection, and unstable rollouts despite small one–step error. **Learned-$J$ (PSD metric).** Here $J$ is *learned* directly from data—i.e., not assembled from the signed incidences $\{D_k\}$ (topology identification dropped)—but is constrained to remain *skew–symmetric* ($J^\top = -J$). The metric is kept positive ($G \succeq 0$). Thus, $J$ no longer reflects mesh topology even though it preserves the algebraic Hamiltonian symmetry. *Expect* stable yet biased dynamics under the *theory* energy: degraded dispersion/momentum behavior and moderate drift. **Learned-$J$ (free metric).** As above, $J$ is *learned* (not incidence–assembled) and remains *skew–symmetric* by construction, but we drop nonnegativity on the learned gains so the induced metric need not be PSD. This preserves the alge-

Table 9: Ablation results. Each number is mean for three seeds.

| Model | One-step MSE | Energy drift | Energy injection | Momentum |
|---|---|---|---|---|
| MeshFT-Net | $\mathbf{4.142 \times 10^{-6}}$ | $\mathbf{2.03 \times 10^{-3}}$ | $\mathbf{8.75 \times 10^{-3}}$ | $4.85 \times 10^{-8}$ |
| No-Orientation $\lvert D_k \rvert$ | $1.08 \times 10^{-4}$ | $13.7$ | $>100$ | $>100$ |
| Scrambled-Topology | $3.85 \times 10^{-5}$ | $13.1$ | $31.1$ | $4.90 \times 10^{-8}$ |
| Indefinite-Metric | $4.145 \times 10^{-6}$ | $2.30 \times 10^{-3}$ | $9.85 \times 10^{-3}$ | $\mathbf{4.28 \times 10^{-8}}$ |
| Learned-$J$ (PSD metric) | $4.37 \times 10^{-6}$ | $1.34 \times 10^{-1}$ | $4.59 \times 10^{-1}$ | $7.79 \times 10^{-3}$ |
| Learned-$J$ (free metric) | $4.16 \times 10^{-6}$ | $8.25$ | $14.8$ | $9.37 \times 10^{-2}$ |

braic skew property of $J$ while violating metric positivity. *Expect* stronger bias, larger energy drift, and more energy injection than in the PSD case.

**Results.** Table 9 reports results of ablation study. Each number is mean for three seeds.

**Discussion.** *Topology (signed $\{D_k\}$) is critical.* Breaking incidence assembly (Scrambled-Topology) or dropping orientations (No-Orientation) yields large energy growth—both drift and injection—even when one-step MSE remains modest. No-Orientation, which violates (O), also shows pronounced momentum blow-up due to loss of action–reaction pairing, confirming that orientation is essential for a skew-symmetric $J$. In addition, Learned-$J$ (PSD metric) is markedly more stable than Learned-$J$ (free metric), yet both underperform the structured MeshFT-Net. Fixing $J$ via the mesh incidences $\{D_k\}$ is essential for stability and fidelity. *Violating metric positivity breaks energy balance.* Indefinite-Metric variants can achieve similar one-step MSE to MeshFT-Net yet inject more energy, underscoring that $J^\top = -J$ (from signed $\{D_k\}$) and a positive energy metric ($G \succeq 0$) must be enforced *together*.

**Takeaway.** These ablations empirically support Theorem 1: the conservative interconnection must be *assembled from the signed incidences $\{D_k\}$* (topology and orientation) and the metric/constitutive block must be *positive-definite*; otherwise long-horizon stability, conservation properties, and physical fidelity deteriorate—even when one-step errors are comparable.

D.8    COMPUTATIONAL COST ANALYSIS

To assess practical efficiency and accuracy, we report mean per-step inference/training latency, peak training CUDA memory, and TSMSE on the analytic wave benchmark (Sec. 5.1), measured on a single NVIDIA H100 under identical configs. These wall-clock numbers are intended as references due to implementation- and kernel-level differences across methods. As summarized in Table 10, across the three resolutions ($32^2, 64^2, 128^2$) MeshFT-Net shows a favorable time/memory/accuracy profile in this setup: it is typically the fastest for inference and training, and its memory footprint is markedly smaller. Latency growth with resolution is also modest for MeshFT-Net, whereas other models increase more noticeably. In these runs, MeshFT-Net achieved lower TSMSE (e.g., $6.5 \times 10^{-5}$ at $128^2$), and it uses substantially fewer parameters. HNN tended to be slower and more memory-intensive on this benchmark.

**Reproducibility Statement**    We took several steps to ensure reproducibility. **Theory:** all assumptions are stated in the main text, and complete proofs of Theorem 1 in Appendix A. **Implementation:** each hyperparameter, training procedures, and evaluation metrics for each experiment are presented in Appendix D. **Execution:** we submit, as *anonymous supplementary material*, an executable package containing scripts that reproduce every result in the paper. Upon acceptance, we will publicly release the full source code, and scripts under an open-source license.

**LLM Usage Disclosure**    We used a general-purpose LLM as an assistive tool for (i) *code design* (module layout, refactoring), (ii) *brainstorming as a discussion partner* (sharpening our research idea), (iii) *paper-writing assistance* (drafting outlines, rephrasing for clarity, grammar/style edits, and figure-caption suggestions), and (iv) *related-work exploration* (flagging potentially missing citations and drafting brief summaries). The LLM's role was strictly assistive: we *always* consulted primary

Table 10: Mean inference and training latency (ms), peak training memory (MB), and TSMSE for four models evaluated on the analytic wave benchmark at three grid resolutions ($32\times32$, $64\times64$, $128\times128$). All values are averages over three random seeds.

| Model | Params | Infer [ms] | Train [ms] | Peak MB (train) | TSMSE |
|---|---|---|---|---|---|
| **32×32** | | | | | |
| MeshFT-Net | 642 | 1.74 | 3.32 | 18.7 | $7.8 \times 10^{-4}$ |
| MGN | 108,930 | 2.58 | 8.07 | 151.9 | $3.5 \times 10^{-1}$ |
| MGN-HP | 217,795 | 2.65 | 26.0 | 462.5 | $3.8 \times 10^{-1}$ |
| HNN | 217,602 | 17.1 | 50.6 | 976.7 | $3.4 \times 10^{-1}$ |
| **64×64** | | | | | |
| MeshFT-Net | 642 | 2.43 | 3.35 | 67.5 | $1.4 \times 10^{-4}$ |
| MGN | 108,930 | 5.48 | 14.4 | 596.5 | $3.0 \times 10^{-1}$ |
| MGN-HP | 217,795 | 5.47 | 45.5 | 1832 | $3.0 \times 10^{-1}$ |
| HNN | 217,602 | 29.9 | 94.4 | 3883 | $3.0 \times 10^{-1}$ |
| **128×128** | | | | | |
| MeshFT-Net | 642 | 2.70 | 3.82 | 262.8 | $6.5 \times 10^{-5}$ |
| MGN | 108,930 | 13.8 | 40.7 | 2379 | $1.2 \times 10^{-1}$ |
| MGN-HP | 217,795 | 13.8 | 142.5 | 7317 | $1.2 \times 10^{-1}$ |
| HNN | 217,602 | 104 | 306.9 | $1.55 \times 10^4$ | $1.2 \times 10^{-1}$ |

sources (original papers, datasets, and official documentation) for any factual or bibliographic claims, and verified all citations. All final technical choices, proofs, claims, and manuscript text were authored and validated by the authors; no outputs were accepted without manual inspection.

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
