# OpenReview forum: "Mesh Field Theory: Port–Hamiltonian Formulation of Mesh-Based Physics"
_ICLR.cc/2026/Conference — Submitted to ICLR 2026_

### Official Review · Reviewer_uYgh · 2025-10-30

**Soundness:** 3
**Presentation:** 4
**Contribution:** 2
**Rating:** 4
**Confidence:** 5

**Summary:**

The authors present a discrete exterior calculus framework for fitting port-Hamiltonian dynamics to co-chain data on meshes. The work is sound and an important direction for the overall scientific machine learning community. These types of formalisms are important to approach meaningful fields. This paper is acceptable for publication assuming the authors address a few major concerns.

My primary feedback is that the authors seem to be unaware of an important body of work where others have already pursued a similar strategy. There are distinct contributions in this paper that merit publication, but the authors must properly contextualize their contribution. There are two primary technical components here that others have also considered: (1) separating the (learnable) metric from the topological exterior derivatives and (2) posing dynamics via a dissipative bracket (port-Hamiltonian in their case) to ensure learned dynamics are stable despite the fact that model form is a priori unknown. The bulk of the literature review is out of date and focuses on the early HNN and LagrangianNN work that looked at low dimensional dynamical systems, and there have been some significant contributions since then which I lay out in the weaknesses section.

From my perspective, the identification of a class of a trainable port-Hamiltonian systems with topological connections is new. To my knowledge this has been done in other dissipative systems but not a port-Hamiltonian.  The authors stress 4 contributions (locally bounded receptive area, permutation symmetry, universality under orientation, and non-increasing energy). The first 3 are good but not particularly novel as they come "for free" working with message passing on graphs. The fourth is a primary contribution.

The benchmarks are of standard quality for ICLR or related AI focused conferences, but weak from the perspective of a serious numerical PDE community. They work through the acoustic scattering problem on the well, but this is simply the linear wave equation and should not be referred to in the title of 5.4 as "real physics field data". It would be much stronger if they attempted a more serious problem with nontrivial physics. At the core of this, there is a very strong assumption in lines 218-221 that J depend only on exterior derivatives independent of state. Very few systems admit J of this form (the wave equation and Maxwell are exceptions, which are coincidentally the cases the authors presented). Therefore it would be much more interesting for the authors to consider one of the significantly more nontrivial physics examples in the well. For example, shear flow could be swapped in with minimal effort and gives a good example of a simple dissipative system.

In the benchmarks, errors are presented in either one-step MSE or energy drift. The is a necessary metric for the method to work, but doesnt capture the primary claimed contribution (stable long time dynamics). Results should be presented for the accuracy of the entire time series.

**Strengths:**

Already covered in my summary. Technically sound and a strong direction of research that many are looking at right now.

The proofs are correct. They could be moved into an appendix as they are more or less a direct consequence of skew/spd symmetries and are standard for port-hamiltonian systems - this is just a suggestion if the authors wanted more space in the manuscript for additional benchmarks.

**Weaknesses:**

The following is a list of modern references from a few groups (Trask, Karniadakis, Cueto, Bronstein) who have worked in this area. Taking a look at these and the papers that cite them will help to contextualize this work.

- The DEC idea was already introduced: https://www.sciencedirect.com/science/article/pii/S0021999122000316
- Using DEC to identify dissipative brackets (although not port-Hamiltonian) https://proceedings.neurips.cc/paper_files/paper/2023/file/7903af0a1cffb43dbb2f8160d110a5f3-Paper-Conference.pdf
----- the authors are likely to find some useful techniques in the appendices that can be used to relax the assumption that J not be state dependent
- Metriplectic brackets (of which port hamiltonians are a special case) are being pursued by several groups:
-- https://arxiv.org/abs/2106.12619
-- https://royalsocietypublishing.org/doi/10.1098/rsta.2021.0207
-- https://arxiv.org/abs/2004.04653
-- https://arxiv.org/abs/2508.12569
- Bronsteins group have presented a number of papers posing GNNs in exterior calculus language: https://arxiv.org/abs/2106.10934

Meshgraphnets and HNNs are not close to state of the art for the well, although they do provide important pedagogical comparisons about whether combining a GNN and hamiltonian dynamics give you something more than the sum of the parts. A comment clarifying this would be useful (autoregressive vision transformers give much more accurate rollouts).

If this work is properly contextualized and the weakness of the benchmarks are improved this is suitable for ICLR - I would bump my score up significantly.

**Questions:**

I had a few technical points that weren't clear to me:

- Why use a splitting scheme to solve this, and not just directly solve the dynamics using a quasi-newton method? These are small 2D systems being considered, so it isn't clear to me that the computational savings justify introducing a splitting error in the dynamics.

---

> ### Author Response · Authors · 2025-11-21
>
> Thank you for the detailed and expert feedback. We respond point by point.
>
> ---
> > My primary feedback is that the authors seem to be unaware of an important body of work where others have already pursued a similar strategy.
>
> We appreciate the references and will recontextualize our contribution accordingly. Our key theoretical distinction is that MeshFT does not predefine a global structure, fixed PDE, or global bracket. Instead, we start from minimal physical principles and prove a local reduction theorem: any mesh dynamics satisfying these principles (Locality, Permutation, Orientation, Energy) reduces to a local port-Hamiltonian form.
>
> We encode this theorem as a structural inductive bias. The mesh topology and orientation fix the interconnection J, while only G and R are learned. Because this constraint is derived from the principles rather than a hand-picked global template, it applies wherever the principles hold and makes the approach equation-agnostic, in contrast to methods that assume a specific global form in advance.
>
> While our work shares the goal of embedding physical structure with the cited studies, our position is distinct:
>
> - Data-driven DEC: These typically fix exterior derivatives and learn metrics within a predefined PDE setting. Our approach does not assume a governing PDE.
>
> - Bracket / Metriplectic / GENERIC methods: Preserve structure by parameterizing each operator, often assuming a specific bracket form. A predefined global template may limit transferability, whereas our principle-level local constraint remains applicable as long as the principles hold.
>
> - GRAND: Views graph networks as discretizations of a diffusion PDE. In contrast, we obtain a local port-Hamiltonian structure from the principles, enforcing only a topology-wired J and learning G and R.
>
> We will make this distinction explicit by citing the all suggested work, adding a short paragraph in Related Works, and noting in Section 3.2 that the reduction is local, principle-driven, and does not assume a global form.
>
> ---
> > should not be referred to in the title of 5.4 as "real physics field data".
>
>  To avoid confusion, we will revise the section title as an “Acoustic Scattering Benchmark” rather than “real physics field data.”
>
> > J depend only on exterior derivatives independent of state.
>
> We agree with that treating J as state-independent is a strong assumption. Our goal in this paper is to introduce and analyze the reduction-based framework and to validate it in a controlled setting; hence we focused on the cleanest “canonical/constant–J” case to make the theoretical ideas transparent. We do not claim that J must be state-independent in general. Appendix B discusses how state-dependent structure maps J(z) should be parameterized in our framework.
>
> > consider one of the significantly more nontrivial physics examples
>
> For a quick response to this point, we added a nonlinear advection–diffusion benchmark. As summarized in Table 5, our approach achieves low trajectory error and exhibits near-zero mass drift without any explicit mass-conservation penalty. This demonstrates that MeshFT’s structure generalizes beyond linear wave equations to a parabolic, weakly nonlinear transport setting. Full details will be provided in Appendix D.6.
>
> ---
> > Results should be presented for the accuracy of the entire time series.
>
> We agree that accuracy over the entire rollout is central to our claim of stable long-time dynamics. Accordingly, our main results now report time-series MSE (TSMSE) in Tables 1, 2, and 4, that is the average squared error over the full rollout.
>
> ---
> > Meshgraphnets and HNNs are not close to state of the art for the well, although they do provide important pedagogical comparisons about whether combining a GNN and hamiltonian dynamics give you something more than the sum of the parts.
>
> Our aim is to propose and test a principle level inductive bias based on the reduction theorem. Thus, we use MGN and HNN as pedagogical and structurally comparable baselines. Also, stronger backbones such as transformers can be combined with our bias. We clarify the rationale for baseline selection in the revised paper. In addition, we added Fourier Neural Operator, GraphCON, and a physics-informed MGN. These additional baselines are reported in Tables 3 and 4.
>
> ---
> >Why use a splitting scheme
>
> We treat the integrator as a design choice. A quasi-Newton or other implicit scheme is feasible, and for the small 2D settings we study the empirical advantage of splitting is modest as you mentioned. The split is not essential to our results, which focus on the structural bias. We will state this explicitly.
>
> ---
> All changes are reflected in the revised manuscript. We appreciate the guidance. With these contextual and empirical refinements, we believe the work clarifies and strengthens its distinct contribution.

---

### Official Review · Reviewer_QG3E · 2025-10-30

**Soundness:** 3
**Presentation:** 3
**Contribution:** 3
**Rating:** 6
**Confidence:** 2

**Summary:**

This paper focuses on Mesh Field Theory (MeshFT) and its neurbl version, MeshFT-Net, which represents an advance in mesh-based continuum physics. Ideas, such as the definition and separation of topological structure from metric and dissipative components, are novel. The authors define and formalize four minimal, yet critical, physical requirements: locality, permutation equivariance, orientation covariance, and energy balance/passivity. They demonstrate that the dynamics of mesh-based physics within these parameters allow local factorization into a port-Hamiltonian structure. The interconnection is determined exclusively by the mesh, leaving only the metric and dissipation components to be learned. Building on this insight, MeshFT-Net hardwires the signed incidence matrix as the fixed interconnection and learns positive-definite metrics and positive-semidefinite dissipative elements. MeshFT-Net is demonstrated to have near-zero energy drift and significantly improved physical fidelity. For the demonstration of this, authors use analytic plane-wave benchmarks, physics-consistency tests, and a real acoustic scattering dataset, "The Well,. Authors compare their approach to  MeshGraphNet (MGN), MGN with a Hamiltonian penalty, and Hamiltonian neural networks.

**Strengths:**

1. Demonstrated that mesh based physics, under simple physical assumptions, reduces to a port-Hamiltonian form where the interconnection is determined solely by mesh topology.
2. MeshFT-Net hard-wires this interconnection and trains exclusively the metric and dissipation terms so that the updates are energy consistent.
3. For 2D wave and acoustic tests, authors demonstrated near zero energy drift and more accurate wave speed and momentum conservation compared to MeshGraphNet and Hamiltonian baseline..
4. Similar or better accuracy is achieved with approximately five times less training data, and generalization occurs for different setups.

**Weaknesses:**

1. Other baselines compared remain classical (MGN, MGN-HP, HNN). Some recent work on graph simulators focus on oversmoothing or long-range dependencies. Additionally, there are some recent variants of MeshGraphNets, such as PI-MGNs (physics-informed MeshGraphNets). These baselines are not included and, as such, makes it harder to assess competitiveness.
2. Authors talk about “mesh-based physics” broadly, but results are mainly on linear wave/acoustics.
3. Probably, the "5x data-efficiency" claim needs additional verification and strict formulation (not rough).

**Questions:**

1. I think authors should narrow the "mesh-based physics" claims or add at least one non-linear / advection example to show the idea generalizes.
2. Make comparison with GraphCON or other recent work focusing to graph simulators with corrected long-range information flow (qualitatively or quantitively). Also, there are some recent variants of MeshGraphNets, such as PI-MGNs (physics-informed MeshGraphNets), it is interesting to compare with them because paper speaks about data efficiency.
3. It is interesting to see some list of counterexamples where main theorem about local reduction doesn't work.
4. As I understand, Figures 2 and 4 provide data-size trends only for the analytic plane-wave task. I have not found it for example for “The Well” dataset. So, please either limit the “5× data efficiency” claim or add train-size analysis for additional datasets.

---

> ### Author Response · Authors · 2025-11-21
>
> Thank you for the thoughtful review, the positive score, and for highlighting both theoretical clarity and empirical strengths. We address each point briefly.
>
> ---
>
> > 1. I think authors should narrow the "mesh-based physics" claims or add at least one non-linear / advection example to show the idea generalizes.
>
> To show the generality of our apporach, we ran a new nonlinear advection–diffusion benchmark. As summarized in Table 5, our approach achieves low trajectory error (TSMSE) and exhibits near-zero mass drift without any explicit mass-conservation constraint. This demonstrates that MeshFT’s structure generalizes beyond linear wave equations to a parabolic, weakly nonlinear transport setting. Full details will be included in the Appendix D.6.
>
> ---
>
> >2. Make comparison with GraphCON or other recent work focusing to graph simulators with corrected long-range information flow (qualitatively or quantitively). Also, there are some recent variants of MeshGraphNets, such as PI-MGNs (physics-informed MeshGraphNets), it is interesting to compare with them because paper speaks about data efficiency.
>
> We added stronger and more diverse baselines under the same training and evaluation protocol. GraphCON, a physics-informed MGN variant, and Fourier Neural Operator are added. We report physics-consistency metrics and OOD metrics. Results support our claims. One-step accuracy is competitive, and our model shows lower energy drift and better physical consistency on long rollouts. These results are included in Table 3 and Table 4 of the revised paper.
>
> ---
>
> >3. It is interesting to see some list of counterexamples where main theorem about local reduction doesn't work.
>
> Counterexamples arise exactly when any principle is violated. Table 9 shows the results of ablation study by modifying model elements that correspond to dropping each principle. Dropping orientation makes J non skew and causes systematic drift and momentum leakage. Scrambling incidences breaks locality and topology, yielding incorrect modal coupling and large long-horizon drift. Allowing an indefinite metric breaks energy balance and produces nonphysical growth. Learning J directly, even if kept skew, breaks the topology link and degrades dispersion and momentum. In this experiments, permutation equivariance is retained. However, in the proof shown in Appendix A, it combined with locality forces incidence-based sparsity and label independence. The proof uses each principle essentially and the derivation fails if any one is removed. Removing any principle breaks the derivation, which matches the observed failures.
>
> The theorem also requires local differentiability in the state. It does not hold on interfaces created by state-dependent discontinuities. For fracture-like phenomena the displacement jumps across the interface. In practice, we encode the loss of interaction as a mesh-topology update to J by removing incidences across the new surface since away from that interface the factorization holds almost everywhere. After the update J preserves incidence sparsity and skew structure on the updated mesh, while G and R continue to be learned locally from geometric and material features. We will add a brief note stating that state-dependent discontinuities break the reduction only on the interface.
>
> ---
>
> >4. So, please either limit the “5× data efficiency” claim or add train-size analysis for additional datasets.
>
> We will restrict the “5×” statement to the analytic plane-wave benchmark where it is measured and avoid implying the same factor elsewhere.
>
> ---
>
> All changes are reflected in the revised manuscript. We appreciate your assessment and believe these directly address competitiveness and evidential clarity while preserving the paper’s contribution.

---

> > ### Comment · Reviewer_QG3E · 2025-11-24
> > **Thank you**
> >
> > Thank you for your detailed response. I am now satisfied with the work.

---

> > > ### Author Response · Authors · 2025-11-24
> > >
> > > Thank you very much for your thoughtful follow-up and for raising the score. I truly appreciate your time and consideration.

---

### Official Review · Reviewer_3vHu · 2025-10-31

**Soundness:** 3
**Presentation:** 3
**Contribution:** 3
**Rating:** 8
**Confidence:** 3

**Summary:**

The paper shows that for mesh-based physics, imposing four minimal physical principles (locality, permutation equivariance, orientation covariance, and energy balance/dissipation inequality) ensures that system's physical wiring is fixed by the mesh itself. Building on this, the authors propose MeshFT-Net, which hard-codes the topology-driven wiring and uses stable time-stepping scheme. On 2D wave tests, out-of-distribution shifts (frequency/resolution/parameters), and a real acoustic dataset MeshFT-Net keeps energy/momentum stable and achieves competitive or better accuracy with less data.

**Strengths:**

- solid theory and result in Theorem 1 shows that mesh-based dynamics of mesh graph nets satisfying the built-in biases (locality and permutation equivariance) together with the physical principles introduced admit a local reduction to port–Hamiltonian representation.

- MeshFT-Net implements the theory in a principled manner

- compelling empirical evidence.

- the method is robust in OOD settings.

- the method is validated on the real physics-field data.

- ablations support the theorem's ingredients.

**Weaknesses:**

- local nature of the theorem: the reduction is jacobian level/local. global guarantees are not analyzed.
- the scope of PDEs used in the main paper are limited to 2D waves and acoustic dataset.
- including neural operator and DEC baselines would strengthen the claims.
- including runtime/wall-clock timings and memory scaling would strengthen the efficiency claims.

**Questions:**

see weaknesses

---

> ### Author Response · Authors · 2025-11-21
>
> Thank you for the positive and constructive review. We respond point by point and keep claims aligned with what we already show.
>
> ---
>
> >local nature of the theorem: the reduction is jacobian level/local. global guarantees are not analyzed.
>
> Thank you for this point. We agree that the reduction is local at the Jacobian level and does not guarantee a global factorization. We will add a brief note to make this explicit not for confusing readers.
>
> In practice this locality is desirable and keeps MeshFT-Net flexible as a consequence. Importantly, we do not assume a single global port-Hamiltonian form, and the reduction applies even when the ground truth is not globally port-Hamiltonian. This provides a principled local inductive bias without imposing a global port-Hamiltonian law. To clarify this point, we will also clarify in Related Works how our approach differs from methods that impose global physical constraints on networks. In addition, Section 4 (l.267-269) presents an optional global parameterization consistent with the reduction, where we define a global dissipative potential to specify the network. More generally, one can use a step-by-step state update based on the local port-Hamiltonian structure without assuming a global form of the dynamics.
>
> ---
>
> > the scope of PDEs used in the main paper are limited to 2D waves and acoustic dataset.
>
> We have added a nonlinear advection–diffusion study in Table 5 to show generality of our approach. These results broaden the empirical scope beyond purely hyperbolic cases and demonstrate that MeshFT's applicability. Although the new task is 2D, the underlying cochain-based reduction theorem is stated and proved for general meshes and is dimension-independent. Thus, extending the approach to 3D is straightforward.
>
> ---
>
> >including neural operator and DEC baselines would strengthen the claims.
>
> We agree that broader baselines strengthen the paper. We added three baselines under the same training and evaluation protocol such as Fourier Neural Operator, GraphCON, and a physics-informed MGN. We report physical consistency metrics and OOD metrics. These results are included in Table 3 and Table 4 of the revised paper. The results support our claim that MeshFT-Net achieve markedly physics-consistency and high OOD performance. For DEC baselines, which build on fixed exterior derivatives and predefined PDE forms, we add a short positioning in Related Works to clarify the difference from our PDE-free approach.
>
> ---
>
> >including runtime/wall-clock timings and memory scaling would strengthen the efficiency claims.
>
> Thank you for this point. I replaced Table 10 in Appendix D.8 with an end-to-end complexity summary of MeshFT-Net.
>
> ---
>
> All changes are reflected in the revised manuscript. We appreciate the positive evaluation and believe these address your concerns while keeping the paper's contribution.

---

### Official Review · Reviewer_qxV9 · 2025-10-31

**Soundness:** 2
**Presentation:** 2
**Contribution:** 3
**Rating:** 4
**Confidence:** 4

**Summary:**

This paper proposes a port-Hamiltonian formulation for mesh-based physics learning. It claims a "local reduction theorem" showing that the mesh dynamics on MeshGraphNet [1] satisfy orientation covariance and energy passivity, and can be factorized into the fixed topological interconnection $J$ (from incidence matrices $D$) and learnable metric/dissipation maps $G$ and $R$. Introducing these physical assumptions in MeshGraphNet, it designs MeshFT-Net, which hardwires the incidence-based skew operator $J$ and learns only symmetric positive (semi-)definite metric maps $(G_\theta, R_\theta)$.

*[1] Pfaff, T., Fortunato, M., Sanchez-Gonzalez, A., & Battaglia, P. (2020, October). Learning mesh-based simulation with graph networks. In International conference on learning representations.*

**Strengths:**

- Theoretical clarity: The separation between topology ($J$) and metric/dissipation ($G$, $R$) is clearly discussed and physically interpretable. In the corresponding model, enforcing skew-symmetry and energy passivity introduces strong structural bias that improve stability but potentially limit flexibility for systems outside the port–Hamiltonian class.

- Readable presentation: Proofs and algorithms are self-contained and systematically laid out.

**Weaknesses:**

See questions.

**Questions:**

- **More experiments are encouraged (e.g., parabolic, elliptic, or nonlinear systems). Or, alternatively, the authors could clarify the constraints under which the model may fail (non-Hamiltonian systems?).** As all experimental data presented here are wave-type systems, it would be helpful to know whether this approach provides benefits on other types of benchmarks.

- **Clarify overall complexity**: Although the authors analyze the complexity of individual modules, an analysis of the full model compared to baselines would give a clearer view of the model’s advantages. This would also help interpret the timing results reported in Table 9 and assess the model’s scalability.

- **Behavior near discontinuities**: The proof relies on local smoothness. How does the model behave near material discontinuities? Is there any demonstration of its ability to handle physical problems with discontinuities? Including benchmarks from the original MeshGraphNet or the Well dataset could be informative.

- **Necessity and minimality of assumptions (L, P, O, E):**

   - Are the assumptions (L, P, O, E) truly minimal for the reduction, or could a weaker or alternative set also yield the factorization? The proofs show sufficiency; can the authors provide evidence of necessity, or counterexamples when any single assumption is removed? If some physical systems slightly violate one of them, the reduction may fail.

- **Uniqueness of the $J$ / $G$ /$R$**:

   * The theorem is stated at the Jacobian level. Is this decomposition unique for nonlinear $F$ globally, or is it only a local Jacobian factorization? (Uniqueness is essential to claim that learning $G$ and $R$ is the only "freedom" left.)

- **Generalization of learned metric/dissipation terms**:
   - Are the learned $(G_\theta, R_\theta)$ mesh-invariant? That is, do they generalize across different embeddings, rescaled edge lengths, or even different mesh topologies? Suggested test: Train on one mesh and test on geometrically deformed or topologically different meshes without retraining.

Overall, I acknowledge that the model is effectively built on a rigorous topological analysis. However, I have reservations about its generalizability across different physical scenarios (*MeshGraphNet currently appears to perform better in this regard*) as well as some theoretical “strengths” and limitations. If the authors can address my concerns without introducing new major issues, I would consider raising my recommendation to marginally accept (readers may interpret my initial overall score as 5).

---

> ### Author Response · Authors · 2025-11-21
>
> Thank you for the careful reading and constructive questions. We address each point concisely below.
>
> ---
> >More experiments are encouraged.
>
> We added a nonlinear advection–diffusion benchmark. As summarized in Table 5, our approach achieves low trajectory error and near-zero mass drift without any explicit mass-conservation constraint. This shows that MeshFT’s structure generalizes beyond linear wave equations to a parabolic, weakly nonlinear transport setting. Full details will be included in Appendix D.6.
>
> >clarify the constraints under which the model may fail (non-Hamiltonian systems?).
>
> The factorization can fail at interfaces for state-dependent discontinuities (e.g., fractures), because the reduction theorem requires the vector field F to be locally differentiable with respect to the state. More details are provided in the section “Behavior near discontinuities.”
>
> ---
> >Clarify overall complexity
>
> We added an end-to-end complexity summary as Table 10. MeshFT-Net uses fixed-width local operators and substantially fewer parameters, which in our measurements yields lower computation time and much lower peak memory while maintaining accuracy. Thus the favorable runtime follows from parameter efficiency. As these results are problem-dependent, we present them as supplemental information in Appendix D.8.
>
> ---
> >Behavior near discontinuities
>
> Our reduction assumes local differentiability in the state. With spatial material discontinuities, the Jacobian still factorizes as (J - R) G. This matches the results shown in Figure 3, where snapshots show that MeshFT-Net handles material discontinuities. When the discontinuity depends on the state, the theorem does not hold on the interface. However, the factorization still holds almost everywhere. For phenomena like fracture, a practical treatment is to encode loss of interaction as a topology change in J by removing incidences across the surface. We will add a brief note that state-dependent discontinuities break the reduction only on the interface.
>
> ---
> >Necessity and minimality of assumptions
>
> The four principles are minimal in practice and theory. Appendix D.6 tests necessity by modifying model elements that correspond to dropping each principle. Removing orientation yields drift and momentum leakage. Scrambling the mesh incidence breaks locality and causes incorrect modal coupling and large long-horizon drift. Also, the proof shown in Appendix A uses each principle essentially and the derivation fails if any one is removed.
>
> ---
> >Uniqueness of the $J$/$R$/$G$
>
> The factorization is local and does not claim a single global nonlinear decomposition. Once the mesh topology is fixed, the interconnection J is determined up to the usual gauge (permutations of cochains and orientation signs). These choices do not affect observables once a convention is fixed. In this sense, the physics fixes J, and only G and R are phenomena-dependent and need be learned. Locally learned factors then assemble under a canonical orientation and Hodge/degree normalization into a globally consistent physical dynamics.
>
> ---
> >Generalization of learned metric/dissipation terms
>
> Table 4 reports out-of-distribution changes in resolution. We observe stable energy and good accuracy. G and R are local geometry-aware maps that take shape features such as edge length adn/or cell area and output positive definite or positive semidefinite blocks. Under relabeling, re-embedding, or rescaling, these inputs transform accordingly, so the same map is expected to yield the correctly scaled metric and dissipation, analogous to mesh-invariant generalization in FEM.
>
> ---
> >potentially limit flexibility for systems outside the port–Hamiltonian class
>
> We appreciate this comment on flexibility. The reduction theorem does not assume a pre-defined global Port-Hamiltonian template. Instead, it proves that any mesh dynamics satisfying the four principles inevitably reduces to this structure locally. The topological wiring J is not a design choice but an intrinsic nature of physics fixed by the principles. This equation-agnostic nature ensures that our framework is not limited to a specific class of equations but generalizes to any system respecting these fundamental principles.
>
> Furthermore, the reduction is local. We do not assume a single global form. An intuitive analogy is the piecewise linear approximation of a nonlinear function. Each segment is linear, yet the function is globally nonlinear. Likewise, local port-Hamiltonian pieces, stitched with a common gauge, define a consistent vector field. This local factorization supports broad flexibility for physical systems while strictly prohibiting non-physical behaviors. We will add a note near Section 3.2 to make this explicit.
>
> ---
> All changes are reflected in the revised manuscript. We hope these clarifications address your concerns. We appreciate your willingness to reconsider toward a marginal accept once these edits are incorporated.

---

> ### Comment · Reviewer_qxV9 · 2025-11-28
>
> I will respond to you one by one.
>
> ---
>
> > have added a nonlinear advection–diffusion benchmark
>
> The added experiment shows a feasible extension of MeshFT to another type of PDE.
>
> However, I observe that some of the advection–diffusion benchmark testing configurations are the same as the previous linear one:
> - Training / Validation: 4000 / 256
> - Time step: smaller than $10^{-3}$ (non-linear advection–diffusion $5 \times 10^{-5}$)
>
> Questions:
> - **Does this model rely on or benefit from a large amount of training data?** Models like MeshGraphNet and FNO have been shown to work on a smaller training dataset size, like 400.
> - **With such a small time-step size, it is unknown how this model perform on practical larger time step / longer time range prediction under the same PDE setting.**
>
>  Can you discuss or show results under smaller training sizes / larger time-step and correspondingly longer time range settings with the same PDE configuration?
>
> > fail at interfaces for state-dependent discontinuities / assumes local differentiability in the state
>
> Good. Please make sure you clearly state this assumption in your future version.
>
> Out of curiosity, how would you extend this method to a non-autonomous equation?
>
> > Clarify overall complexity
>
> I suggest including a theoretical discussion of the computational complexity to help readers understand the cost of each part of the model design.
>
> > Necessity and minimality of assumptions
>
> **Between lines x–x? I can not find them in the current version.**
>
> > In this sense, the physics fixes J, and only G and R are phenomena-dependent and need be learned.
>
> Well, do you have any assumptions that guide the choice of a “better” G, and R? Or does the choice of G and R depends on initialization and optimization?
>
> > Table 4 reports out-of-distribution changes in resolution
>
> Good. If experiments similar to MeshGraphNet (like Deforming Plate) could be conducted, that would be amazing. If not, it is okay.
>
> > potentially limit flexibility for systems outside the port–Hamiltonian class
>
> The answer is satisfactory. You should highlight the class of systems covered by your model in the future versions.
>
> It is more like a model composed of certain types of Lego-like sub-blocks, and the final shape classes depend on, but are limited by, these basic blocks. Rather than being a general approximator, it focuses on a certain class of PDEs. Analogous to FEM or spectral methods, behavior outside their “comfort zone” is not the scope of this method.
>
> In summary, this work presents a well-established topological framework and has evaluated the feasibility of this method. Now I still keep the score of 5 (Positive for my preference of the theoretical parts, Negative for my concern on the practical performance). If authors could reduce **my highlighted concerns**, I will raise my score to 6 (Since soundness will be improved).

---

> ### Author Response · Authors · 2025-11-28
>
> Thank you for the thoughtful follow-up. We address each point and will reflect them in the manuscript.
>
> ---
>
> > **Can you discuss or show results under smaller training sizes / larger time-step and correspondingly longer time range settings with the same PDE configuration?**
>
> We extended the experiments by sweeping both the training data size and the physical time step \\(\Delta t\\) for the same weakly nonlinear advection–diffusion configuration, reporting results for training sizes \\(400,4000\\) and \\(\Delta t\in\\{5\times10^{-5},10^{-3}\\}\\) which is shown in **Table 5**. The results indicate that more data helps robustness, especially when temporal sampling is coarse (e.g., reducing Mass Drift), but the model’s stability and predictive accuracy are not contingent on having a huge dataset. Even with only 400 samples, the prediction error (TSMSE) remains low and comparable to the large-data regime.
>
> In revisiting this benchmark to ensure rigor, we realized that our earlier use of a logistic reaction term was inappropriate for the autonomous setting; we now use the corrected backbone \\(\frac{\partial{q}}{\partial{t}}+\nabla\cdot\big((a+\beta q)q\big)=\kappa\Delta q\\) for all runs. Additionally, once a minor implementation detail was fixed, the previously reported single- and two-stage comparison and \\(K\\)-step free-rollout ablation no longer showed a meaningful qualitative gap, so we removed them and focused on the more interpretable \\((\Delta t,\text{ train size})\\) sweep. We appreciate that the reviewer’s comment led to a cleaner and more informative experiment.
>
> > Please make sure you clearly state this assumption in your future version.
>
> We will state explicitly that the theorem requires local differentiability in the state and fails on the discontinuity interface.
>
> > How would you extend the method to non-autonomous equations?
>
> Time dependence can enter through \\(G\\) and \\(R\\), written as \\(G(z,t)\\) and \\(R(z,t)\\), and external inputs can be modeled as source terms (ports) in the port-Hamiltonian framework.
>
> > suggest including a theoretical discussion of the computational complexity.
>
> We will include a short theoretical cost decomposition.
>
> Let $N_0, N_1, N_2, N_3$ be the numbers of 0-, 1-, 2-, and 3-cochains (vertices, edges, faces, cells), with $N_3 = 0$ in 2D. Let $\text{nnz}(J)$ be the number of nonzeros in the sparse interconnection $J$ built from all cochain adjacencies.
>
> * Assembly: Building the incidence and assembling $J$ costs $O(N_0 + N_1 + N_2 + N_3)$ once.
> * Evaluation: Evaluating local maps for $G$ and $R$ costs $O(N_0 + N_1 + N_2 + N_3)$ per step.
> * Time-stepping: Sparse matrix-vector products with $J$ plus block-diagonal operations $G$ and $R$ cost $O(\mathrm{nnz}(J)+N_0+N_1+N_2+N_3)$ per step.
> * Memory: The memory requirement is $O(\text{nnz}(J) + N_0 + N_1 + N_2 + N_3)$.
>
> On well-shaped meshes, the local valence is bounded, so $\text{nnz}(J)$ grows proportionally to the mesh size. Hence, the overall cost is near-linear in mesh size.
>
> > **Between lines x–x? I can not find them in the current version.**
>
> Apologies, the earlier cross-reference was incorrect. The ablations that mirror dropping each principle are in **Appendix D.7 (lines 1154 to 1218)**. The full proof appears in **lines 540 to 761**.
>
> > Well, do you have any assumptions that guide the choice of a “better” G, and R? Or does the choice of G and R depends on initialization and optimization?
>
> The reduction theorem ensures that, locally, \\(\partial F/\partial z = (J − R) G\\), so suitable \\(G\\) and \\(R\\) exist. Choosing them is a learning problem and depends on parameterization, initialization, and optimization. We enforce SPD for \\(G\\) and PSD for \\(R\\) by construction (e.g., small positive-diagonal initialization and softplus parameterization). Designing better objectives and optimizers for \\(G\\) and \\(R\\) is problem dependent and is an interesting direction for future work.
>
> > If experiments similar to MeshGraphNet (like Deforming Plate) could be conducted, that would be amazing. If not, it is okay.
>
> Thank you for the suggestion. We will not include a deforming-plate benchmark in this revision.
>
> > You should highlight the class of systems covered by your model in the future versions.
>
> We will explicitly state the class of systems covered by MeshFT-Net.
>
> ---
> We hope these updates address your concerns. I truly appreciate your time and consideration.

---

### Author Response · Authors · 2025-11-29
**Summary of Revisions on Experiments, Baselines, and Theoretical Context**

We present this summary to assist the Area Chair in assessing our revisions. We sincerely thank all reviewers for their time and constructive feedback. We appreciate the detailed assessment of our theoretical approach and the recognition of this research direction's relevance to the scientific machine learning community.

In response to the reviews, we have substantially extended the experiments and clarified the theoretical scope as requested. We also note that reviewer QG3E has already indicated that the revision satisfactorily addresses their main concerns. Reviewer qxV9 likewise acknowledged our theoretical contribution, while keeping a score of 5 due to concerns about practical performance, and explicitly stated that they would raise their score to 6 if their highlighted concerns were reduced. These concerns were already addressed in our response and are further targeted by the revisions below, in particular the expanded nonlinear advection–diffusion experiments.

---
**1. Validation on Nonlinear Advection–Diffusion Systems (Addressing all reviewers' concerns for experimental scope)**

To address the concern that our experiments were limited to linear waves, we added a **nonlinear advection–diffusion benchmark** (Table 5).
* **Result:** Our method maintains low trajectory error and near-zero mass drift without explicit constraints. This confirms that our principle-driven inductive bias is robust even for **parabolic and weakly nonlinear dynamics**.
---
**2. Comparison with Modern Baselines (Addressing 3vHu, QG3E, uYgh)**

We have expanded comparisons to include broader baselines: **Fourier Neural Operator (FNO)**, **GraphCON**, and **PI-MGN**.
* **Result:** As shown in revised Tables 3 and 4, MeshFT-Net demonstrates competitive accuracy and **improved physical consistency** (lower energy/momentum drift) compared to these strong baselines, reinforcing our claims on physical consistency and OOD robustness.
---
**3. Clarification of Theoretical Context (Addressing uYgh)**

Following the expert feedback from uYgh, we have rewritten the Related Works to explicitly contextualize our work alongside **Data-driven DEC, GENERIC, and Metriplectic frameworks**.
* **Distinction:** We clarified that unlike approaches assuming a global pre-defined structure, MeshFT derives a *local* Port-Hamiltonian structure directly from minimal physical principles (Locality, Permutation, Orientation, Energy).
---
**4. Expressivity of the Local Reduction (Addressing qxV9, 3vHu)**

We addressed the concern that the Port-Hamiltonian assumption might limit the model's flexibility. We clarified that our reduction theorem does not impose a rigid global template; rather, it proves that *any* mesh dynamics satisfying the four principles inevitably reduces to this structure locally. This ensures the model remains **equation-agnostic and flexible** while strictly prohibiting non-physical behaviors.

---
We have also included **Time-Series MSE (TSMSE)** for long-term stability analysis and an **end-to-end complexity analysis** (Appendix D.8). Overall, the revised manuscript offers a more comprehensive empirical evaluation and a clearer theoretical narrative, strengthening MeshFT-Net as a framework that bridges geometric principles and deep learning for physical mesh dynamics.

---

### Meta-Review · Area_Chair_dnME · 2026-01-02

**Summary:**

This paper introduces a new port-Hamiltonian formulation for mesh-based continuum physics. One reviewer has concerns about the novelty of the proposed method because the method utilizes a similar strategy as previous works. Therefore, the contribution of the paper should be re-stated. Moreover, although authors have included some experiments to address the concerns in the rebuttal, some detailed comparisons with extensive baselines under the same configuration are still not clear. Therefore, I believe the paper is not ready for publication.

**Reviewer Concerns:**

See above.

**Reviewer Scores:**

The two reviewers with negative scores could maintain the score.

---

### Decision · Program_Chairs · 2026-01-26

Reject